# UI-Genie: A Self-Improving Approach for Iteratively Boosting MLLM-based Mobile GUI Agents

**Han Xiao**[1,2†]**, Guozhi Wang**[2]**, Yuxiang Chai**[1,2†]**, Zimu Lu**[1]**, Weifeng Lin**[1,2†]**,**
**Hao He**[1]**, Lue Fan**[1]**, Liuyang Bian**[2]**, Rui Hu**[2]**, Liang Liu**[2]**,**
**Shuai Ren**[2‡✉]**, Yafei Wen**[2]**, Xiaoxin Chen**[2]**, Aojun Zhou**[1✉]**, Hongsheng Li**[1,3,4✉]

[1]CUHK MMLab    [2]vivo AI Lab    [3]Shanghai AI Laboratory    [4]Ace Robotics
{1155229123@link,hsli@ee,aojunzhou@link}.cuhk.edu.hk
shuai.ren@vivo.com
✉Corresponding author    ‡Project lead    †Interns at vivo.

## Abstract

In this paper, we introduce UI-Genie, a self-improving framework addressing two key challenges in GUI agents: verification of trajectory outcome is challenging and high-quality training data are not scalable. These challenges are addressed by a reward model and a self-improving pipeline, respectively. The reward model, UI-Genie-RM, features an image-text interleaved architecture that efficiently processes historical context and unifies action-level and task-level rewards. To support the training of UI-Genie-RM, we develop deliberately-designed data generation strategies including rule-based verification, controlled trajectory corruption, and hard negative mining. To address the second challenge, a self-improvement pipeline progressively expands solvable complex GUI tasks by enhancing both the agent and reward models through reward-guided exploration and outcome verification in dynamic environments. For training the model, we generate UI-Genie-RM-517k and UI-Genie-Agent-16k, establishing the first reward-specific dataset for GUI agents while demonstrating high-quality synthetic trajectory generation without manual annotation. Experimental results show that UI-Genie achieves state-of-the-art performance across multiple GUI agent benchmarks with three generations of data-model self-improvement. We open-source our complete framework implementation and generated datasets to facilitate further research in https://github.com/Euphoria16/UI-Genie.

## 1 Introduction

Large Language Models (LLMs) [23, 24, 1] have demonstrated remarkable potential for powering autonomous mobile GUI agents capable of completing complex tasks through natural language instructions. The emergence of Multi-modal Large Language Models (MLLMs) [13, 17, 2] represents a significant advancement in this field, introducing enhanced perception and reasoning abilities [46, 8] crucial for interface navigation. By directly perceiving screenshots, identifying functional UI elements, and generating executable action sequences in a vision-centric manner, MLLM-based GUI agents have achieved better performance compared to their text-only counterparts.

While recent advancements in general MLLMs have demonstrated the effectiveness of large-scale synthetic data generation [3, 52], creating high-quality synthetic data for GUI agents remains particularly challenging: *(1) Accurate Trajectory Outcome Verification.* Verifying GUI agent trajectory outcomes presents unique challenges that distinguish it from other multi-modal understanding tasks. Unlike common question-answering tasks where correctness can be directly judged by checking a final answer, the completion state of GUI agent tasks is heavily dependent on historical context,

Table 1: UI-Genie dataset statistics. UI-Genie-RM-517k is the first dedicated GUI agent reward dataset, while UI-Genie-Agent-16k contains synthetic trajectories without manual annotation.

| Reward Datasets | Size | Manual Annotation | Data Source | Positive Num | Negative Num |
|---|---|---|---|---|---|
| UI-Genie-RM-517k | 263k 170k 24k 59k | × | AndroidControl AMEX AndroidLab Exploration | 121k 68k 14k 29k | 142k 102k 10k 30k |
| Agent Datasets | Size | Manual Annotation | Average Steps | Trajectories | Task Instruct |
| Android Control | 88k | ✓ | 5.5 | 15283 | High&Low |
| AMEX | 34k | ✓ | 12.8 | 2946 | High |
| AndroidLab | 6k | ✓ | 8.6 | 726 | High |
| UI-Genie-Agent-16k | 16k | × | 7.1 | 2208 | High&Low |

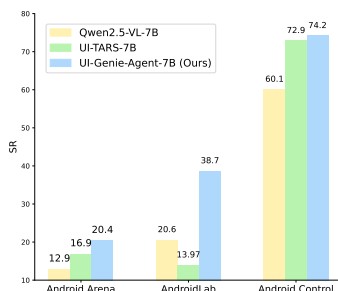

Figure 1: Performance comparison between UI-Genie, Qwen2.5-VL and UI-TARS on three benchmarks.

making evaluation substantially more complex. Existing assessment frameworks, including those utilizing proprietary models as judges [25, 43], fail to provide accurate outcome verification and reliable step-level validation of intermediate actions. *(2) Scalable High-Quality Training Data.* Due to the lack of reliable verification methods, current training approaches still rely on human-annotated operation trajectories [15, 5, 44], which are inherently time-consuming, expensive, and difficult to scale. These manually-created datasets suffer from limited volume and diversity, particularly lacking high-quality demonstrations for complex multi-step tasks.

To address these challenges, we introduce **UI-Genie**, a self-improving framework that generates high-quality synthetic trajectories with accurate process supervision, eliminating the need for extensive human annotation. To provide fine-grained trajectory assessment, we build UI-Genie-RM, the first reward model specifically designed for evaluating GUI agents. Our reward model features an image-text interleaved architecture that efficiently processes historical screenshots and actions as context while unifying both action-level and task-level rewards. This innovative approach enables both single-step and multi-step trajectory evaluation within the same architecture. To overcome the significant challenge of the absence of training datasets designed for GUI agent reward models, we develop deliberate data generation strategies, including rule-based verification, controlled trajectory corruption, and hard negative mining based on open-source training datasets. This comprehensive dataset construction process, combined with UI-Genie-RM's long-context processing capabilities and unified reward representation, enables accurate validation at both the action level and task level without requiring manual annotation.

To generate high-quality synthetic trajectories, we leverage UI-Genie-RM for process supervision during agent exploration in dynamic environments. In these rollouts, UI-Genie-RM ranks candidate actions by reward scores and expands only the most promising trajectories. At terminal states, we use outcome verification to assess trajectory success. To address low initial success rates on complex tasks, we introduce a self-improvement framework where both the agent and reward models evolve iteratively. Specifically, we expand training data with successful trajectories to boost performance on increasingly complex tasks, as well as correctly labeled steps from failed trajectories to refine the reward model. Our framework overcomes the limitations of static, manually-annotated datasets and enables continuous improvement in GUI agent performance. Through this process, we generate UI-Genie-RM-517k, the first large-scale reward dataset for GUI agents, and UI-Genie-Agent-16k, a high-quality synthetic trajectory dataset. As shown in Tab. 1, our method yields substantially more data without human annotation. Experiments show that UI-Genie achieves state-of-the-art results with three self-improvement cycles, as illustrated in Fig. 1. We release our full open-source framework to support further research in GUI agents.

Our contributions can be summarized as follows:

1. We develop UI-Genie-RM, a specialized reward model for GUI trajectory assessment with rich interleaved image-text observations, while unifying action-level and task-level rewards.

2. We introduce UI-Genie, a novel self-improving framework where both the agent model and reward model evolve iteratively. This framework progressively solves complex GUI tasks through reward-guided trajectory exploration, training data expansion and iterative model fine-tuning.

3. We create and open-source two novel datasets (UI-Genie-RM-517k and UI-Genie-Agent-16k) along with our complete framework implementation, establishing the first reward-specific dataset for GUI agents and demonstrating synthetic trajectory generation without manual annotation.

## 2 Related Work

**Multi-modal GUI Agents.** Recent advances in Large Language Models (LLMs) [23, 24, 28, 38] have enabled the development of GUI agents, with Multi-modal LLMs (MLLMs) further enhancing UI information perception [9, 12]. Prior work [1, 36, 37, 50] leveraging commercial MLLMs has shown promising results across mobile and desktop environments. Meanwhile, open-source approaches have focused on specialized architectures and training methods. OS-Atlas [42] adopts a unified action space for different UI grounding tasks. UI-TARS [27] introduces comprehensive pretraining on element grounding and reasoning. [20, 19] apply rule-based Reinforcement Learning [11] and GRPO [30] to achieve competitive performance. Despite these advances, a significant challenge persists: most approaches remain dependent on human-annotated trajectories. While some methods [44, 33, 27, 22, 34] incorporate synthetic trajectory generation, they still rely on human effort or closed API models for verification, limiting scalability. In contrast, our work enables fully automatic trajectory synthesis and evaluation with an effective self-improvement pipeline.

**Reward Models.** Reward models have been widely used in reasoning-intensive tasks to assist in test-time scaling [32, 16, 47], expand solution paths [10], and facilitate model fine-tuning [35, 39, 48, 14]. Prior studies such as [16] and [35] demonstrate that, compared to outcome-supervision reward models (ORMs), process-supervision reward models (PRMs) can better perceive the correctness of intermediate steps in reasoning trajectories, thereby providing more reliable feedback. To avoid the high cost of step-level human annotation, various works [21, 39] have proposed methods to automatically obtain process supervision by estimating the potential of intermediate steps to lead to the correct final result. Similar to these studies, our work also leverages a PRM trained on step-level rewards obtained through a sophisticated rollout in dynamic environments. Recent works [40, 49, 41] have extended the application of PRMs from LLMs to MLLMs. [33] employ GPT-4o as a trajectory-level reward model to assess the quality of GUI agent trajectories. Similarly, [26] trains an outcome reward model for trajectories, enabling the model to learn from both successful and failed attempts. In contrast, our method creates a large-scale reward training dataset from scratch and does not rely on costly proprietary models. Moreover, unlike these outcome-only rewards, our model provides both task-level and action-level rewards and refines both reward and agent models iteratively with the increasingly complex training data.

## 3 Method

This section introduces UI-Genie, a framework for solving mobile GUI agent challenges through an iterative self-improving approach. We begin by formulating the reward modeling task in Sec. 3.1. Sec. 3.2 presents the specialized reward model. Then, we detail the dataset construction strategies in Sec. 3.3. Finally, Sec. 3.4 explains how the self-improvement pipeline works to continuously enhance both components.

### 3.1 Task Formulation for GUI Trajectory Reward

**Action-level reward.** Given a task goal $\mathcal{G}$, current screenshot $\mathcal{I}_t$, and historical observations $\mathcal{O}_t = (\mathcal{I}_0, \mathcal{A}_0), (\mathcal{I}_1, \mathcal{A}_1), ..., (\mathcal{I}_{t-1}, \mathcal{A}_{t-1})$, the objective is to evaluate whether a candidate action $\mathcal{A}_t$ constitutes a correct step toward achieving $\mathcal{G}$. The reward model outputs a positive reward ($y = 1$) for correct actions and negative reward ($y = 0$) otherwise.

**Task-level reward.** This task evaluates whether an entire trajectory successfully completes the goal $\mathcal{G}$. The input consists of all actions and resulting screenshots in the trajectory, and the output is a single reward score indicating overall success or failure.

### 3.2 UI-Genie-RM: A Unified Reward Model

Previous approaches of trajectory verification [27, 42, 43, 25] rely on human efforts or proprietary models for validation, which are inherently costly and difficult to scale. Moreover, these methods fails to provide validation of intermediate steps, constraining their practical adaptability. To address these limitations, we develop UI-Genie-RM, a GUI reward model which builds upon a standard MLLM backbone, but incorporates crucial modifications to handle sequential GUI interactions and unify step-level and outcome-level rewards.

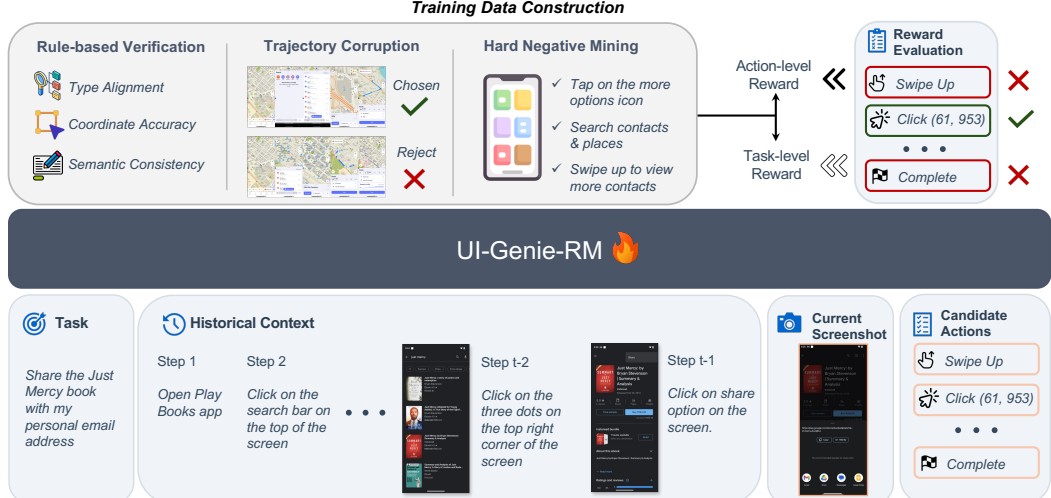

Figure 2: **Overview of UI-Genie-RM model and reward training data construction.** The model processes task instruction, historical context, current screenshot, and candidate action as inputs. Outputs are supervised by both action-level and task-level rewards. The training data are constructed by rule-based verification, trajectory corruption, and hard negative mining processes.

**Model Architecture.** UI-Genie-RM determines if a proposed action $\mathcal{A}_t$ advances the task goal $\mathcal{G}$, given the current state. Because GUI tasks depend on long-term context, accurate action assessment requires understanding more than the current screenshot. For example, in a cross-app task where a user copies a headline from News to Notes, the correct action in Notes (typing the headline) depends on knowing what was seen earlier in News, not just the present screenshot.

To address this issue, UI-Genie-RM uses an image-text interleaved architecture that leverages past interactions to evaluate current actions. As shown in Fig. 2, the model takes in the task goal, current screenshot, and candidate actions, plus an interleaved sequence of recent screenshots and action history for context. This helps the model track progress and understand the impact of previous steps.

Processing full histories is costly for complex tasks. We address this by a context window using only the five most recent screenshots, while summarizing earlier actions as language descriptions. Our agent predicts these natural language summaries (e.g., "Access Categories & Budget section") alongside action primitives (e.g., `click(x,y)`), providing semantic context even when screenshots are missing. This representation balances historical coverage and computational efficiency.

**Unified Action-Level and Task-Level Rewards.** Our model architecture's ability to process historical information provides necessary context for trajectory-level judgments, which enables us to unify step-level and task-level assessment within the same architecture. To achieve this goal, we formulate "`task completion`" as a special action within the action space, allowing us to judge whether this terminal action is appropriate given all past GUI interactions when evaluating task-level outcomes.

This unified representation allows both action-level and task-level rewards to be handled through the same optimization objective, eliminating the need for separate models across different evaluation granularities. Specifically, our reward model assigns binary rewards to both actions and tasks given the task goal $\mathcal{G}$, current screenshot $\mathcal{I}_t$, and historical context descriptions $\mathcal{H} = \{\mathcal{H}_0, \mathcal{H}_1, ..., \mathcal{H}_{t-1}\}$. We define the reward ground truth $\mathcal{R}_t \in \{0, 1\}$, where 1 indicates a correct action and 0 indicates an incorrect one. Following previous work [51], we formulate reward prediction as a next-token generation task. The model is trained to maximize the likelihood of predicting $y^+$ for correct actions and $y^-$ for incorrect ones. Given a dataset $\mathcal{D}$ containing positive action examples $\mathcal{A}_t^+$ and negative examples $\mathcal{A}_t^-$, our training objective minimizes

$$\mathcal{L}_{\text{RM}} = - \sum_{(\mathcal{A}_t^+, y^+) \in \mathcal{D}} \log P(y^+ | \mathcal{G}, \mathcal{I}_t, \mathcal{H}, \mathcal{A}_t^+) - \sum_{(\mathcal{A}_t^-, y^-) \in \mathcal{D}} \log P(y^- | \mathcal{G}, \mathcal{I}_t, \mathcal{H}, \mathcal{A}_t^-), \quad (1)$$

where $y^+$ and $y^-$ denote the positive and negative action labels, respectively.

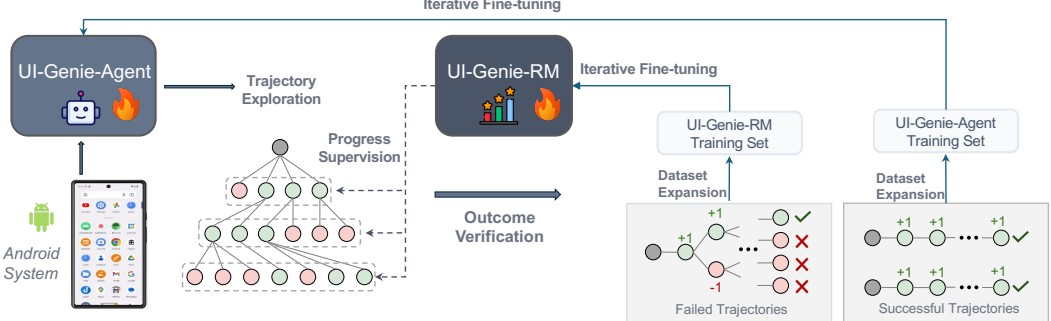

Figure 3: **Self-improvement of agent and reward models for UI-Genie.** It expands training sets for both agent and reward models through reward-guided trajectory exploration and outcome verification, then finetunes both models. This process repeats iteratively to improve capabilities on increasingly complex tasks.

## 3.3 Data Construction for Reward Model Training

To train an MLLM into an effective GUI agent reward model, we develop deliberately-designed data construction strategies that create a comprehensive training corpus of approximately 517k reward data. The detailed data statistics are presented in Tab. 1. We initially leverage task instructions and corresponding ground-truth actions from open-source datasets [15, 5, 44] as sources. We then apply three complementary data generation techniques on the base data source to create positive and negative samples, resulting in 458k synthetic training samples for our initial reward model. This dataset is further enriched with an additional 59k samples acquired through trajectory exploration in dynamic environments as part of our self-improvement pipeline described in Sec. 3.4.

**Rule-based Verification.** We first train an initial agent model on the base GUI operation trajectory datasets [15, 5, 44]. For each task instruction, we generate candidate actions by sampling prediction from this initial agent given the same task instructions, then validate them against ground truth actions using three criteria: (1) *Type alignment*—whether predicted and ground truth actions must share the same type; (2) *Coordinate accuracy*—for spatial operations like "click" and "long press", predicted coordinates must fall within ground truth bounding boxes to ensure functional validity; (3) *Semantic consistency*—for text-based operations like "typing", generated content must maintain semantic equivalence with ground truth.

**Controlled Trajectory Corruption.** To create negative trajectory-level samples, we systematically perturb successful trajectories through three mechanisms: (1) *Early truncation*—terminating trajectories at intermediate steps to simulate incomplete executions; (2) *Cross-task substitution*—replacing trajectory segments with actions from different tasks within the same application, mimicking goal misinterpretation; (3) *Redundant continuation*—appending unnecessary steps beyond task completion to model failure in recognizing terminal states.

**Hard Negative Mining.** After obtaining the initial agent model (trained on existing trajectory datasets) and reward model (whose training dataset is constructed with rule-based verification and controlled trajectory corruption techniques), we adopt a hard-negative mining process to find challenging negative examples in reward dataset. We start with existing annotations from the AMEX dataset [5], which provides target tasks, ground-truth actions and descriptions for other target-unrelated UI elements. We then use an open-source LLM [18] to convert these descriptions into executable actions. All of these actions are regarded as step-level negative actions for the target task. After that, we use the initial reward model to score them. If the reward model mistakenly assigns a positive score to one action, this action is identified as hard negative.

These complementary strategies yield a diverse, high-quality dataset that enables effective training of UI-Genie-RM across varying difficulty levels and failure modes, establishing a robust foundation for accurate single-step and multi-step assessment.

## 3.4 Iterative Self-Improvement of Agent and Reward Models

Following the training of initial UI-Genie-RM reward model, we establish a self-improvement framework that dynamically generates GUI operation trajectories and progressively enhances both reward and agent models. As illustrated in Fig. 3, this framework operates within a dynamic Android

environment where agent actions directly modify environmental states. We first introduce our approach of reward-guided trajectory exploration and training data expansion, then present our recipe of iterative model fine-tuning with increased task complexity.

**Reward-Guided Trajectory Exploration.** For GUI trajectory exploration, we find traditional Monte Carlo Tree Search (MCTS) mechanisms [4, 31] are suboptimal due to their computational expense. More importantly, conventional MCTS fails to effectively address the unique characteristics of GUI interaction spaces. Specifically, unlike mathematical problem solving where erroneous steps lead to incorrect answers, GUI interactions often include invalid actions that maintain the last state (e.g., clicking blank areas or non-functional elements). To address these challenges, we develop a reward-guided beam search approach inspired by [7]. At each step, UI-Genie-Agent generates ten candidate actions that are evaluated by UI-Genie-RM to produce step-level rewards. We rank these candidates by cumulative rewards along each partial trajectory and retain only the top-5 paths, as indicated by the green circles in the tree in Fig. 3. This exploration efficiently discovers both successful completion paths and informative partial trajectories for our training data expansion process.

**Training Data Expansion with Outcome Verification.** Our framework iteratively expands training data for both the agent and reward models through a dual labeling approach. To expand the training set of UI-Genie-Agent, we apply outcome verification using UI-Genie-RM to identify successful completions. These successful trajectories are directly added to the agent's training set to enhance its generalization capabilities. To expand the training set of UI-Genie-RM, we employ a potential-based labeling approach [39] for step-level reward annotation. For successful trajectories, each constituent step is annotated as correct (label $y^+$). For unsuccessful trajectories, we determine the viability of each intermediate step through continuation rollouts. From each intermediate step, we sample five continuation paths and evaluate their outcomes. If any of these rollouts successfully completes the task, we classify the original step as correct ($y^+$), indicating it maintains task completion potential despite being part of a failed trajectory. This approach automatically generates fine-grained step-level supervision for refining the reward model.

**Iterative Model Fine-tuning.** We implement our framework through three iterative rounds with gradually increasing task complexity: **1.** Initial generation: We utilize task instructions directly derived from source training data [44, 5], establishing baseline performance. **2.** Expansion generation: We introduce novel task instructions generated by prompting an advanced open-source LLM [18] with app descriptions and example tasks from the first generation. **3.** Complex task generation: We combine filtered instructions of failed tasks from previous rounds with manually crafted complex scenarios requiring more than ten steps to complete.

This progressive approach creates a positive feedback loop between framework components: enhanced agent capabilities generate more diverse, successful trajectories for increasingly complex tasks; these trajectories provide richer supervision signals for the reward model; the refined reward model delivers more precise guidance during exploration; and this improved guidance enables the discovery of solutions to even more complex tasks. Each generation thus reinforces both components' capabilities while progressively expanding the frontier of solvable GUI tasks.

## 4 Experiments

In this section, we evaluate UI-Genie across diverse benchmarks designed to assess both GUI agent capabilities and reward model accuracy. We first describe our implementation details, followed by a comprehensive overview of the evaluation benchmarks and metrics. Ablation studies are provided in Sec. A to validate the design choices of each component.

### 4.1 Implementation Details

We implement both UI-Genie-RM and UI-Genie-Agent based on the Qwen2.5-VL family of models due to its strong multimodal understanding capabilities. For the reward model, we adopt Qwen2.5-VL-7B as the backbone. For agent models, we experiment with three variants across different model sizes: 3B, 7B, and 72B, enabling analysis of scaling effects on performance.

**Reward Model Training.** UI-Genie-RM is initialized with Qwen2.5-VL-7B and trained using a binary classification objective. We introduce special tokens $\langle|+|\rangle$ and $\langle|-|\rangle$ to represent positive and negative class labels, respectively, and formulate reward prediction as a next-token generation

Table 2: Performance comparison on AndroidControl benchmark [15]. The table presents commercial systems (Claude Computer Use and GPT-4o), open-source foundation models, and our approach. Higher values are better for all metrics: SR (Success Rate), Type (Action Type Accuracy), and Grounding (UI Element Interaction Accuracy) across both low-level and high-level task settings. Highest values are indicated as **bold**.

| Agent | Model Size | AndroidControl-Low | | | AndroidControl-High | | |
|---|---|---|---|---|---|---|---|
| | | Type | Grounding | SR | Type | Grounding | SR |
| Claude Computer Use | – | 74.3 | 0.0 | 19.4 | 63.7 | 0.0 | 12.5 |
| GPT-4o | – | 74.3 | 0.0 | 19.4 | 66.3 | 0.0 | 20.8 |
| Aria-UI | 3.9B | – | 87.7 | 67.3 | – | 43.2 | 10.2 |
| UI-R1 | 3B | 94.3 | 82.6 | – | – | – | – |
| GUI-R1 | 3B | – | – | – | 58.0 | 56.2 | 46.6 |
| Qwen2.5-VL | 3B | 79.3 | 72.3 | 90.8 | – | – | 63.7 |
| OS-Atlas | 4B | 91.9 | 83.8 | 80.6 | 84.7 | 73.8 | 67.5 |
| UI-TARS | 2B | **98.1** | 87.3 | 89.3 | 81.2 | 78.4 | 68.9 |
| InfiGUI-R1 | 3B | 96.0 | 93.2 | 92.1 | **82.7** | 74.4 | 71.1 |
| **UI-Genie-Agent (Ours)** | 3B | 97.8 | **94.7** | **93.8** | 82.5 | **82.5** | **72.9** |
| OS-Genesis | 7B | 90.7 | – | 74.2 | 66.2 | – | 44.5 |
| GUI-R1 | 7B | – | – | – | 71.6 | 65.6 | 51.7 |
| SeeClick | 9.6B | 93.0 | 73.4 | 75.0 | 82.9 | 62.9 | 59.1 |
| Qwen2.5-VL | 7B | – | – | 91.4 | – | – | 60.1 |
| Aguvis | 7B | – | – | 80.5 | – | – | 61.5 |
| OS-Atlas | 7B | 93.6 | 88.0 | 85.2 | 85.2 | 78.5 | 71.2 |
| UI-TARS | 7B | 98.0 | 89.3 | 90.8 | **83.7** | 80.5 | 72.5 |
| **UI-Genie-Agent (Ours)** | 7B | **98.1** | **94.9** | **94.3** | 83.5 | **82.9** | **74.2** |
| Aguvis | 72B | – | – | 84.4 | – | – | 66.4 |
| Qwen2.5-VL | 72B | – | – | 93.7 | – | – | 67.4 |
| UI-TARS | 72B | 98.1 | 89.9 | 91.3 | 85.2 | 81.5 | 74.7 |
| **UI-Genie-Agent (Ours)** | 72B | **98.3** | **95.4** | **94.8** | **84.9** | **86.3** | **77.0** |

task. We first conduct supervised fine-tuning on our constructed dataset of 458k samples derived from existing datasets. Subsequently, we perform iterative refinement using a total of 59k process-reward samples collected during self-improvement cycles. We train the model using the AdamW optimizer with a learning rate of 1e-5 and a global batch size of 160.

**Agent Model Training.** We first establish an initial agent model based on Qwen2.5-VL-7B using existing datasets including AndroidControl [15], AMEX [5], and AndroidLab [44]. This initial model performs trajectory exploration in our dynamic Android environment under UI-Genie-RM's supervision. Through three rounds of iterative improvement, we generate UI-Genie-Agent-16k, a synthetic dataset containing 16k high-quality trajectories without manual annotation. For variants beyond 7B, we train UI-Genie-Agent-3B and UI-Genie-Agent-72B using UI-Genie-Agent-16k and the aforementioned open-source datasets for one epoch. All agent models are fine-tuned with learning rate 1e-5 and batch size 160.

## 4.2 Evaluation Benchmarks

### 4.2.1 Agent Model Evaluation

To comprehensively evaluate UI-Genie, we conduct experiments on both *static and dynamic benchmarks*. Static evaluation assesses the agent's ability to predict correct actions in static environments where only a single frame with ground-truth action history is input, while dynamic evaluation tests the agent's performance in dynamic, interactive settings in an emulator where each action directly affects task progress and completion.

**Static Evaluation.** We utilize the AndroidControl [15] benchmark to evaluate UI-Genie-Agent's planning and action-execution capabilities in static mobile environments without actually executing actions on a device. This benchmark compares the agent's predicted actions with ground truth actions, providing a controlled and reproducible evaluation framework.

Following previous work [45, 27], we evaluate our agent on two settings: (1) high-level tasks requiring the agent to autonomously plan and execute actions to complete a task goal based on screenshots and

Table 3: Performance comparison on AndroidLab [44]. The table presents commercial systems (top section), open-source foundation models and their fine-tuned variants (middle section), and our approach (bottom). The best performance values are highlighted in **bold**, while underlined values denote the sub-optimal results.

| Agent | Model Size | Sub-Goal Success Rate | Reversed Redundancy Ratio | Reasonable Operation Ratio | Success Rate |
|---|---|---|---|---|---|
| Gemini-1.0 | – | 12.6 | 72.5 | 76.7 | 10.9 |
| Claude-3-Opus | – | 15.1 | 81.4 | 83.9 | 13.0 |
| Gemini-1.5-Pro | – | 18.5 | 106.0 | **91.5** | 16.7 |
| Claude-3.5-Sonnet | – | 32.7 | **113.4** | 81.2 | 29.0 |
| GPT-4o | – | **35.0** | 87.3 | 85.4 | 31.2 |
| AutoGLM | – | – | – | – | **36.2** |
| LLaMA3.2-11B-Vision | 11B | 13.0 | 61.7 | 87.9 | 10.1 |
| CogVLM2-ft | 19B | 16.1 | 57.4 | 85.6 | 11.6 |
| UI-TARS-ft | 7B | 18.2 | 64.1 | 88.0 | 14.0 |
| Qwen2.5-VL | 7B | 18.7 | 70.6 | 76.8 | 14.9 |
| Qwen2-VL-ft | 7B | 22.6 | 65.2 | 88.3 | 18.1 |
| Qwen2.5-VL-ft | 7B | 24.2 | 59.5 | 76.3 | 20.6 |
| **UI-Genie-Agent (Ours)** | 3B | 35.4 | 91.3 | 90.6 | 28.8 |
| **UI-Genie-Agent (Ours)** | 7B | **46.3** | **92.8** | **91.4** | **38.7** |
| UI-TARS | 72B | 12.0 | 40.8 | 70.0 | 9.6 |
| UI-TARS-ft | 72B | 28.4 | 81.4 | 81.6 | 22.1 |
| Qwen2.5-VL | 72B | 26.1 | 68.7 | 81.4 | 23.9 |
| Qwen2-VL-ft | 72B | 29.3 | 84.5 | 90.2 | 24.6 |
| Qwen2.5-VL-ft | 72B | 30.9 | 81.3 | 79.3 | 25.0 |
| **UI-Genie-Agent (Ours)** | 72B | **48.2** | **92.1** | **93.3** | **41.2** |

action history; and (2) low-level tasks that provide step-by-step low-level instructions for the agent to execute predefined actions. For both settings, we employ three primary metrics: (1) Success Rate (SR), measuring the percentage of successfully completed actions; (2) Type Accuracy, assessing the agent's ability to correctly predict action types (e.g., click, long-press, scroll); and (3) Grounding Accuracy, evaluating the agent's precision in locating and interacting with the correct UI elements.

Tab. 2 presents a comparison of our UI-Genie-Agent against proprietary systems and open-source foundation models on the AndroidControl benchmark [15]. Our approach consistently outperforms existing methods across all model sizes. The 3B variant achieves 93.8% and 72.9% success rates on low and high-level tasks respectively, surpassing previous SOTA by 1.7% and 1.8%. The 7B and 72B variants further improve performance, with our 72B model reaching 77.0% success rate on high-level tasks, exceeding UI-TARS by 2.3%. The consistent improvements across model sizes demonstrate the robust scalability of our UI-Genie framework with the integration of high-quality trajectories during agent training.

**Dynamic Evaluation.** AndroidLab [44] provides a controlled emulator environment where the agent's actions are directly executed, and success depends on its ability to adapt to dynamic UI states resulting from its interactions. The benchmark comprises 138 tasks across nine categories of frequently used offline static applications (e.g., maps, calendars, books, music players, etc.). These tasks simulate typical daily user interactions such as managing events, editing notes, and retrieving information.

We evaluate performance on AndroidLab using the metrics defined in [44], which capture task completion at different levels: (1) Sub-Goal Success Rate (Sub-SR), evaluating completion of individual task components; (2) Reversed Redundancy Ratio (RRR), assessing operational efficiency compared to human benchmarks; (3) Reasonable Operation Ratio (ROR), measuring the proportion of effective operations that change the screen state; and (4) Success Rate (SR), measuring the percentage of tasks where all sub-goals are completed. As shown in Tab. 3, our approach demonstrates superior performance compared to both commercial systems and various open-source foundation models and their fine-tuned variants.

Besides, Android Agent Arena (A3) [6] presents a more comprehensive and challenging online evaluation environment. It employs an actual Android device emulator with 201 tasks spanning 20 predefined online top-rated applications selected from 18 categories (news, travel, ticketing, etc.). This benchmark includes particularly demanding tasks requiring more than 15 interaction steps and multi-frame information retrieval. These features make A3 highly effective for assessing an agent's ability to perform tasks in real-life scenarios and execute complex action sequences in response to evolving environments. The metrics detailed in A3 [6] are used to evaluate UI-Genie's

Table 4: Evaluation results on the A3 benchmark [6]. Func SR: Task Success Rate by Evaluation Function; LLM SR: Task Success Rate by Commercial LLM Evaluation; EASR: Essential State Achievement Rate. Highest overall scores are indicated as **bold** for Func SR, underlined for LLM SR, and in *italic* for EASR. Op., S.Q. and M.Q. represent operation, single-frame query and multi-frame query in A3 respectively.

| Agent | Metric | Easy | Med. | Hard | Op. | S. Q. | M. Q. | Overall |
|---|---|---|---|---|---|---|---|---|
| Claude-3.5-sonnet | Func SR | 11.7 | 2.6 | 0.0 | 8.4 | 2.0 | 0.0 | 6.5 |
| | LLM SR | 13.8 | 2.6 | 0.0 | 9.8 | 2.0 | 0.0 | 8.8 |
| | ESAR | 23.6 | 16.7 | 14.2 | 22.1 | 14.7 | 14.9 | 17.6 |
| Qwen2.5-VL-7B | Func SR | 23.4 | 5.2 | 0.0 | 17.5 | 0.0 | 0.0 | 12.9 |
| | LLM SR | 27.7 | 7.8 | 0.0 | 19.6 | 8.2 | 0.0 | 15.9 |
| | ESAR | 39.8 | 27.2 | 29.4 | 36.4 | 25.6 | 28.3 | 33.2 |
| UI-TARS-7B | Func SR | 28.7 | 9.1 | 0.0 | 21.0 | 0.0 | 0.0 | 16.9 |
| | LLM SR | 34.0 | 15.6 | 0.0 | 23.8 | 16.3 | 0.0 | 21.9 |
| | ESAR | 55.8 | 40.1 | 41.8 | 51.1 | 36.2 | 35.7 | 46.5 |
| **UI-Genie-Agent-7B (Ours)** | Func SR | 38.3 | 6.5 | 0.0 | 28.0 | 2.1 | 0.0 | **20.4** |
| | LLM SR | 40.4 | 14.3 | 0.0 | 30.1 | 12.2 | 0.0 | 24.4 |
| | ESAR | 61.8 | 40.4 | 43.6 | 57.8 | 39.5 | 34.9 | *51.4* |

capability. This evaluation incorporates two distinct methods for assessing task completion: (i) function-based evaluation and (ii) commercial LLM-based essential states evaluation. Employing these different approaches provides a comprehensive understanding of UI-Genie's performance from multiple perspectives.

Tab. 4 demonstrates UI-Genie's superior performance compared to previous SOTA agents such as UI-TARS. Specifically, UI-Genie achieves an approximate 5% higher success rate in function-based evaluation and a 3% improvement in commercial LLM-based evaluation. Furthermore, its Essential State Achievement Rate (ESAR) surpasses that of UI-TARS by approximately 5% overall, highlighting UI-Genie's enhanced capabilities in real-life scenarios.

### 4.2.2 Reward Model Evaluation

We conduct a evaluation of UI-Genie-RM using a custom benchmark, since there is no established standard benchmark for GUI agent reward models. Our benchmark derives from test sets of three open-source datasets: AndroidControl [15], AMEX [5], and AndroidLab [44]. For step-level evaluation, we sample 200 distinct ground truth actions as positive examples from each dataset, pairing each with a corresponding negative action generated by the agent model and verified through rule-based methods. For outcome-level evaluation, we include 200 ground truth trajectories as positive samples, complemented by an equal number of negative trajectories created through controlled trajectory corruption for AndroidControl and AMEX. We further augment this with 100 additional trajectories (50 successful, 50 failed) generated during AndroidLab dynamic testing and validated using pre-defined rules. This creates a comprehensive benchmark containing 1,050 paired items.

We evaluate UI-Genie-RM's accuracy using F1-score for both step-level and outcome-level reward assessment, with detailed performance breakdowns across task complexity categories: easy (under 5 steps), medium (5-10 steps), and hard (over 10 steps). For comparison, we establish strong baselines using advanced proprietary MLLMs (GPT-4o, Gemini1.5-pro, Gemini2.5-pro) and open-source models (Qwen2.5-VL-7B/72B). As these models lack specific reward modeling training, we carefully craft prompts to adapt them for this purpose. Additionally, we enhance inputs with visual prompts in screenshots to compensate for these models' limitations in processing spatial coordinates.

Tab. 5 presents our evaluation results. UI-Genie-RM consistently outperforms all baseline models across both evaluation types and all task complexity levels. The performance gap becomes particularly pronounced in hard tasks, where UI-Genie-RM maintains robust performance (68.7% F1-score for step-level, 70.5% for outcome-level) while other models show significant degradation. This demonstrates the effectiveness of our specialized architecture and training approach, particularly for complex GUI interactions requiring extensive historical context understanding.

## 5 Discussions

We presented UI-Genie, a self-improving framework that addresses key challenges in training GUI agents by generating high-quality synthetic trajectories and introducing a specialized reward model,

Table 5: Performance comparison of reward models on step-level and outcome-level evaluation across different task complexity categories. We report F1-score (%) for classifying actions and outcomes as positive or negative.

| Model | Step Reward | | | | Outcome Reward | | | |
|---|---|---|---|---|---|---|---|---|
| | Overall | Easy | Medium | Hard | Overall | Easy | Medium | Hard |
| GPT-4o | 68.1 | 72.5 | 65.2 | 54.8 | 66.7 | 70.3 | 58.8 | 56.8 |
| Gemini1.5-pro | 65.1 | 70.5 | 62.5 | 51.2 | 71.5 | 72.4 | 67.5 | 54.5 |
| Gemini2.5-pro | 72.9 | 74.1 | 64.4 | 55.8 | 74.3 | 78.2 | 73.2 | 64.9 |
| Qwen2.5-VL-7B | 56.6 | 60.2 | 55.1 | 47.9 | 58.9 | 64.1 | 48.4 | 41.3 |
| Qwen2.5-VL-72B | 66.2 | 71.5 | 64.5 | 50.9 | 67.6 | 68.6 | 65.5 | 46.4 |
| **UI-Genie-RM (Ours)** | **79.6** | **83.5** | **81.9** | **68.7** | **82.1** | **84.3** | **81.3** | **70.5** |

UI-Genie-RM. Our framework eliminates reliance on manual annotation through iterative agent-reward model co-evolution, achieving state-of-the-art performance and producing two novel datasets. The results demonstrate the potential of synthetic data generation with process supervision for scalable GUI agent development.

**Limitations.** While effective overall, our reward model may occasionally generate suboptimal rewards signals, resulting in failed trajectories during training data expansion. Although we observe significant performance improvements when applying our synthetic data to enhance agent capabilities, the framework cannot guarantee the production of fully correct trajectories across all GUI tasks.

**Broader Impact.** This work can help people with disabilities better use mobile devices, making the technology more accessible to a broader population. However, the high computation cost of training such a model results in significant carbon emission. Moreover, while the UI-Genie framework aims to advance automation in GUI interactions, we acknowledge the importance of recognizing potential risks associated with fully automated bots. If not properly monitored, these bots could lead to misuse, such as conducting financial fraud or stealing private data without authorization.

# 6 Acknowledgment

This study was supported in part by National Key R&D Program of China Project 2022ZD0161100, in part by the Centre for Perceptual and Interactive Intelligence, a CUHK-led InnoCentre under the InnoHK initiative of the Innovation and Technology Commission of the Hong Kong Special Administrative Region Government, in part by NSFC-RGC Project N_CUHK498/24, and in part by Guangdong Basic and Applied Basic Research Foundation (No. 2023B1515130008, XW).

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

# A  Ablation Study

## A.1  UI-Genie-RM for Test-time Scaling.

Beyond serving as an effective model for synthetic GUI operation trajectory selection, UI-Genie-RM can also enhance agent model performance through a best-of-N sampling strategy. During inference, the agent model generate N candidate actions, which are then ranked by UI-Genie-RM to select the highest-scoring one as the final output.

We evaluate this approach on the AndroidControl benchmark using Qwen2.5-VL base models. Our experiments measure both step-level success rates (accuracy of predicted single-step actions compared to ground truth) and task-level success rates (successful completion of entire tasks, where each step must align with ground truth). Additionally, we categorize the results by task difficulty: easy (less than 5 steps), medium (5-10 steps), and difficult tasks (more than 10 steps).

As shown in Tab. 6, UI-Genie-RM consistently improves performance across both metrics and all model sizes. The Qwen2.5-VL-3B model shows substantial gains, with the step-level success rate increasing from 60.3% to 63.1% and the task-level success rate improving from 7.5% to 10.2% when using best-of-10 sampling. Similarly, the Qwen2.5-VL-7B model demonstrates improvements, particularly for medium-difficulty tasks where the task-level success rate increases from 10.5% to 12.2%. Importantly, our results demonstrate scaling benefits when expanding the sampling space from N=5 to N=10. This performance improvement indicates that UI-Genie-RM effectively identifies optimal actions from larger candidate pools, showcasing its ability to provide accurate reward signals for GUI interactions across varying task complexities.

## A.2  Effectiveness of UI-Genie-RM Architecture

To evaluate the sufficiency of using five most recent screenshots to process historical context for UI-Genie-RM, we conduct an ablation study analyzing our model's performance and efficiency across different task complexities.

For simple tasks (<5 steps), the actual input is naturally limited by the number of all available history (e.g., 3 screenshots at step 4). We evaluate preserving at most 1, 3, and 5 screenshots in Tab. 7. While using one screenshot is most efficient, it shows limited performance due to insufficient historical context, particularly in outcome reward evaluation. Using up to five screenshots achieves the best performance without introducing significant inference overhead, since the actual number of processed images remains low.

For complex tasks ($\geq$ 8 steps), we evaluate the screenshot number at 5, 8, and 10. We observe that using 5 screenshots achieve comparable performance to 8 screenshots, while increasing to 10 leads to performance degradation, as presented in Tab. 7. This suggests that older screenshots beyond the 5 most recent ones may distract the model by less relevant information. Moreover, larger window sizes significantly increase the computational cost. These results demonstrate the effectiveness of our choice of using five most recent screenshots to achieve the best accuracy and efficiency trade-off.

## A.3  Effectiveness of Self-Improvement Approach

To quantitatively evaluate our iterative self-improvement framework, we track the performance enhancement of both the agent and reward models across multiple rounds. Fig. 4 illustrates the progressive performance improvement measured through task success rate on AndroidLab (for UI-Genie-Agent-7B) and step reward accuracy (for UI-Genie-RM). Our results demonstrate substantial and consistent improvements across iterations. Notably, the task success rate of UI-Genie-Agent-7B on AndroidLab increases substantially from an initial 18.1% (round 0) to 38.7% (round 3), finally surpassing the state-of-the-art AutoGLM method. Concurrently, the reward model accuracy of UI-Genie-RM improves from 68.2% to 79.6%, demonstrating the effectiveness of our self-improvement approach. The most dramatic improvement occurs during the first round, highlighting the strength of our reward-guided exploration for trajectory discovery. Rounds 2 and 3 show continued improvement as the framework progressively incorporates more complex tasks and refines both models. These results validate our self-improvement framework's ability to break through the initial performance limitations on complex tasks through mutual enhancement of agent and reward models.

Table 6: Performance comparison of Qwen2.5-VL models with and without UI-Genie-RM test-time scaling. Results show step-level success rate (Step SR) and task-level success rate (Task SR).

| Model | N | Overall | | Easy | | Medium | | Hard | |
|---|---|---|---|---|---|---|---|---|---|
| | | Step SR | Task SR | Step SR | Task SR | Step SR | Task SR | Step SR | Task SR |
| QwenVL2.5-3B | - | 60.3 | 7.5 | 50.4 | 11.6 | 63.5 | 7.1 | 58.3 | 0.9 |
| +UI-Genie-RM | 5 | 61.8 | 8.3 | 52.2 | 13.5 | 65.7 | 7.6 | 58.4 | 0.9 |
| +UI-Genie-RM | 10 | **63.1** | **10.2** | **53.5** | **15.5** | **67.0** | **9.7** | **59.6** | **1.9** |
| QwenVL2.5-7B | - | 64.9 | 14.7 | 64.4 | 29.7 | 66.7 | 10.5 | 61.1 | 1.4 |
| +UI-Genie-RM | 5 | 64.9 | 15.1 | **65.8** | 31.1 | 67.6 | 10.7 | 58.6 | 0.9 |
| +UI-Genie-RM | 10 | **65.2** | **16.2** | 65.5 | **31.7** | **68.1** | **12.2** | **58.9** | 1.4 |

Table 7: Reward model performance with different seen images.

| Task Complexity | Img Num | Step-level Acc(%) | Outcome Acc(%) | Inference Time(s/iter) |
|---|---|---|---|---|
| Simple | 1 | 76.4 | 74.5 | 0.59 |
| | 3 | 78.2 | 79.8 | 1.05 |
| | 5 | 83.5 | 84.3 | 1.57 |
| Complex | 5 | 72.9 | 73.5 | 2.46 |
| | 8 | 73.4 | 75.2 | 5.10 |
| | 10 | 69.7 | 71.3 | 7.81 |

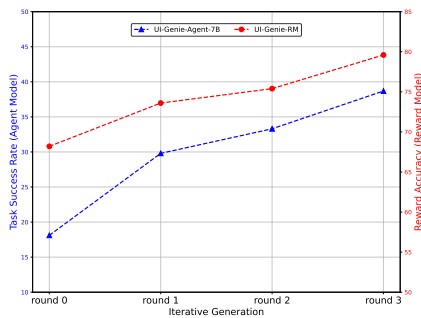

Figure 4: Performance evolution across iterative self-improvement rounds.

## A.4 Ablation on Complex Task Generation and Reward Model Updating

To evaluate the effectiveness of each component of our method, we conduct ablations on the following two aspects, as shown in Tab. 8: **(1) Removing complex task generation.** Without introducing challenging tasks in the final round, the agent's improvement is limited after Round 2 (31.2 to 32.6), highlighting the importance of a curriculum of increasing difficulty. **(2) Freezing reward model updates.** Updating the reward model during the self-improvement phase provides essential guidance for trajectory exploration, resulting in higher-quality synthetic data generation and enhanced performance of both models.

# B Dataset Analysis

We provide the detailed statistics of the UI-Genie-Agent-16k dataset to analyze its diversity.

**Action Distribution:** As shown in Tab. 9, our dataset contains 8 distinct action types with a realistic distribution of GUI interactions. While Click is naturally the most frequent action, the significant presence of Type (9.7%) and Swipe (6.7%) ensures the agent learns to handle complex data entry and navigation tasks.

Table 8: More ablation studies.

| Complex Task Generation | Reward Model Update | Reward Accuracy | | | | Agent Success Rate | | | |
|---|---|---|---|---|---|---|---|---|---|
| | | Round 0 | Round 1 | Round 2 | Round 3 | Round 0 | Round 1 | Round 2 | Round 3 |
| × | × | 69.2 | 69.2 | 69.2 | 69.2 | 18.1 | 29.8 | 31.2 | 32.6 |
| × | ✓ | 69.2 | 73.6 | 75.4 | 77.1 | 18.1 | 29.8 | 33.3 | 34.1 |
| ✓ | × | 69.2 | 69.2 | 69.2 | 69.2 | 18.1 | 29.8 | 31.2 | 35.5 |
| ✓ | ✓ | 69.2 | 73.6 | 75.4 | 79.6 | 18.1 | 29.8 | 33.3 | 38.7 |

**Goal Diversity:** We categorize task goals into 7 distinct categories in Tab. 10, showing significant diversity across mobile GUI scenarios. The task goals span a wide range of realistic user intents, preventing model overfitting to a narrow set of tasks.

**Task Complexity:** Crucially, our dataset emphasizes challenging tasks over trivial simple tasks, as presented in Tab. 11. We find that medium (5-10 steps) and hard tasks (>10 steps) comprise over 91% of the dataset, which is essential for developing highly capable agents.

Table 9: Dataset analysis on action distribution.

| Action Type | Open | Click | Swipe | Long press | Type | System button | Wait | Terminate |
|---|---|---|---|---|---|---|---|---|
| Number | 2208 | 8428 | 1054 | 15 | 1526 | 218 | 25 | 2208 |
| Percentage | 14.08% | 53.74% | 6.72% | 0.10% | 9.73% | 1.39% | 0.15% | 14.08% |

Table 10: Dataset analysis on goal category.

| Goal Category | Number | Percentage |
|---|---|---|
| Configuration & Settings | 518 | 23.46% |
| Information Retrieval | 610 | 27.63% |
| Task & Item Management | 291 | 13.18% |
| Search Tasks | 194 | 8.79% |
| Application Control | 178 | 8.06% |
| Site Navigation | 183 | 8.29% |
| Data Entry | 234 | 10.60% |

Table 11: Dataset analysis on task difficulty.

| Task Difficulty | Trajectory Length | Number | Percentage |
|---|---|---|---|
| Easy | <5 Steps | 182 | 8.24% |
| Medium | 5-10 Steps | 1165 | 52.76% |
| Hard | >10 Steps | 861 | 38.99% |

## C   More Evaluation Results

### C.1   Performance on AndroidWorld Benchmark

We evaluate our final agent models on the AndroidWorld benchmark. As shown in the Tab. 12, UI-Genie-Agent-7B achieves a success rate of 36.2%, surpassing strong proprietary models like GPT-4o. Our 72B model shows significant performance gains compared to its base model Qwen2.5-VL-72B, outperforming all other methods.

### C.2   Out-of-distribution Test of UI-Genie-RM

We evaluate UI-Genie-RM on an out-of-distribution test set constructed from the AndroidWorld [29] benchmark, using 11 apps that were unseen during UI-Genie-RM training. As shown in Tab. 13, UI-Genie-RM achieves performance competitive with the powerful proprietary model Gemini2.5-pro and significantly outperforms other open-source MLLMs, demonstrating strong generalization to unseen domain.

## D   UI-Genie Training

### D.1   Definition of Action Space

To facilitate effective cross-dataset training, UI-Genie implements a comprehensive unified action space across all datasets including AndroidControl [15], AMEX [5], AndroidLab [44], and our newly-generated synthetic data. While building upon the foundation established by Qwen2.5-VL models [3], we incorporate several critical enhancements to improve operational flexibility and generalization capabilities. As detailed in Tab. 14, our action space comprises eight fundamental action types: open,

Table 12: Comparisons on AndroidWorld benchmark.

| Models | Agent Success Rate (%) |
|---|---|
| GPT-4o | 34.5 |
| Claude Computer Use | 27.9 |
| Qwen2.5-VL-7B | 22.0 |
| UI-TARS-7B | 33.0 |
| **UI-Genie-Agent-7B** | **36.2** |
| Qwen2.5-VL-72B | 35.0 |
| Aguvis-72B | 26.1 |
| **UI-Genie-Agent-72B** | **47.4** |

Table 13: OOD test of UI-Genie-RM.

| Model | Step-level Acc (%) | Outcome Acc (%) |
|---|---|---|
| Gemini2.5-pro | 71.5 | 75.7 |
| Qwen2.5-VL-7B | 46.8 | 64.4 |
| Qwen2.5-VL-72B | 54.2 | 68.0 |
| **UI-Genie-RM** | 74.5 | 76.2 |

`click`, `swipe`, `long_press`, `type`, `system_button`, `wait`, and `terminate`. For each action type, we define specific parameters that ensure precise execution and compatibility with real-world mobile environments.

A key advancement over the original Qwen2.5-VL implementation is UI-Genie's dual-mode interaction capability. Beyond predicting absolute coordinates, our model can process Set-of-Mark (SoM) annotations, where interactive UI elements are labeled with numeric tags in the screenshot. This enables UI-Genie to predict element indices rather than exact coordinates when provided Set-of-Mark visual prompts, significantly enhancing generalization across diverse interface layouts. For each action type, we define tailored parameters: the `open` action specifies an app name; `click` and `long_press` use either coordinates or SoM indices to identify targets; `swipe` includes starting position (coordinate/SoM), direction (up/down/left/right), and distance (short/medium/long); `wait` specifies duration in seconds; and `terminate` indicates task completion status (success/failure). Additionally, we incorporate a semantic "action_desc" parameter across all action types, offering crucial semantics when incorporated into the historical record of operations. Moreover, in the negative sample construction process, we utilize the sampled action and its action_desc to evaluate the action alignment effectively. We use some pre-defined rule-based functions and potentially trigger an LLM to judge whether the sampled action is a true negative or equivalent to positive.

Table 14: Action space and parameter specification for UI-Genie.

| Action Type | Parameters |
|---|---|
| `open` | text; action_desc |
| `click` | coordinate; som; action_desc |
| `swipe` | coordinate; som; direction; distance; action_desc |
| `long_press` | coordinate; som; action_desc |
| `type` | text; action_desc |
| `system_button` | button; action_desc |
| `wait` | time; action_desc |
| `terminate` | status; action_desc |

## D.2 Training details

We implement both UI-Genie-RM and UI-Genie-Agent based on the Qwen2.5-VL family of models due to its strong multimodal understanding capabilities. For the reward model, we adopt Qwen2.5-VL-7B as the backbone. For agent models, we experiment with three variants across different model sizes: 3B, 7B, and 72B, enabling analysis of scaling effects on performance.

**Reward Model Training.** For training UI-Genie-RM, we adopt Qwen2.5-VL-7B as the backbone. We introduce special tokens $\langle |+| \rangle$ and $\langle |-| \rangle$ into the original vocabulary of Qwen2.5-VL to represent

You are an expert in evaluating the performance of a phone operating agent.
The agent is designed to help a user to complete a task or retrieve information from the phone.
Given the user's task instruction, the agent's history of actions and current action, your goal is to decide whether the agent's current action is correct or not. Each action in the sequence is preceded by a corresponding screenshot that captures the context in which the action occurs. If you see a green hollow circle surrounded by a red rectangle with a "P" in the top-right corner, this is an additional marker I've added to indicate where the action is performed on the screen.

##Evaluation Criteria
Whether the agent's current action is correct and corresponding to the user's task instruction

##Instructions
1. Review the task instruction and the history of actions leading up to the current step.
2. Check if the current action logically progresses toward the goal (e.g., opening the right app, entering correct input, navigating efficiently).
3. If the action is irrelevant, incorrect, or leads the agent away from the goal, mark it as [wrong].
4. If the action is appropriate and contributes to task completion, mark it as [correct].
5. If the agent is stuck in a loop (repeating the same action without progress), mark it as [wrong].

##IMPORTANT
1. An action always follows a corresponding screenshot (even if only the last few are provided).
2. You should whether answer [correct] or [wrong].

##Input
The goal of the task: {task_instruction}
history of actions: {formatted_actions}
current action of the agent: {current_action}

Figure 5: Step-level reward evaluation prompt used for comparative baseline models.

positive and negative class labels, respectively, thus formulating reward prediction as a next-token generation task. We first conduct supervised fine-tuning on our constructed dataset of 458k samples derived from existing datasets. Subsequently, we perform iterative refinement using a total of 59k process-reward samples collected during self-improvement cycles. After each cycle, we train the model using the AdamW optimizer with a learning rate of 1e-5 and a global batch size of 160 on 20 L40s machines with 4 GPUs each. The model parameters are fully finetuned, while we keep the vision encoder frozen. The input image is limited to have a maximum pixels of 602112.

**Agent Model Training.** We first establish an initial agent model based on Qwen2.5-VL-7B using existing datasets including AndroidControl [15], AMEX [5], and AndroidLab [44]. This initial model is fully finetuned on these datasets with only freezing the vision encoder, with learning rate 1e-5 and batch size 160.

Then we use this initial agent model for trajectory exploration in our dynamic Android environment under UI-Genie-RM's supervision. Through three rounds of iterative improvement, we generate UI-Genie-Agent-16k, a synthetic dataset containing 16k high-quality trajectories without manual annotation. After each round, we continue fine-tuning this 7B model using newly collected operational trajectories with the same training hyperparameters for one epoch.

Furthermore, we train UI-Genie-Agent-3B and UI-Genie-Agent-72B on UI-Genie-Agent-16k and the aforementioned open-source datasets for one epoch. For the 3B model, it is fully finetuned with vision encoder parameter frozen using the AdamW optimizer with a learning rate of 1e-5 and a global batch size of 160. For the 72B model, we finetune it use lora due to GPU memory limit. Specifically, we use rslora and we add lora modules to the vision encoder, projection layer and the LLM. The lora rank and lora alpha are set to be 256 respectively, and we use PiSSA method to initialize the lora weights. All experiments for training agent models are conducted on 20 L40s machines with 4 GPUs each.

You are an expert in evaluating the performance of a phone operating agent.
The agent is designed to help a user to complete a task or retrieve information from the phone.
Given the user's task instruction, the agent's trajectory of actions and final response, your goal is to decide whether the agent's execution is successful or not. Each action in the sequence is preceded by a corresponding screenshot that captures the context in which the action occurs. If you see a green hollow circle surrounded by a red rectangle with a "P" in the top-right corner, this is an additional marker I've added to indicate where the action is performed on the screen.
##Evaluation Criteria
Whether the agent's trajectory is effective and corresponding to the user's task instruction
##Instructions
 1. Review the agent's actions and the corresponding screenshoot step by step.
 2. if the agent is stuck in the very first login stage, which means it fails to log into target website at the beginning, that's a failure.
 3. Determine if the agent has achieved all the requirements of the task instruction based on the trajectory.
 4. The agent sometimes can't stop after finishing a task and continue doing repeated actions. This should be considered failed as the agent should output 'finished' at the final step.
 5. If the agent is stuck in the loop, which means they don't even get close to the goal before they get stuck in the loop, that's a failure.
 6. If the task is to retrieve information or answer a question, the agent can be considered successful only when it provide the correct and necessary information. You should carefully check if the provided answer is correct and sufficient to answer the user query.
##IMPORTANT
 1. In the trajectory, an action always follows a corresponding screenshot in the images(only several last screenshots are provided, but the trajectory of actions is compactly presented in this context), which shows the observation of the agent.
 2. You should whether answer [success] or [failure].
##Input
 The goal of the task: {task_instruction}
 trajectory of actions: {formatted_actions}
 final response of the agent: {final_response}

Figure 6: Outcome-level reward evaluation prompt used for comparative baseline models.

## D.3 Implementation Details

We implement both UI-Genie-RM and UI-Genie-Agent based on the Qwen2.5-VL model family, leveraging its strong multimodal understanding capabilities. For our experiments, we develop variants across three model sizes (3B, 7B, and 72B) to analyze performance scaling effects. All experiments are conducted on a distributed training setup of 20 L40s machines with 4 GPUs each.

### D.3.1 Reward Model Training

For UI-Genie-RM, we adopt Qwen2.5-VL-7B as the backbone and introduce special tokens $\langle | + | \rangle$ and $\langle | - | \rangle$ into the vocabulary to represent positive and negative class labels, effectively formulating reward prediction as a next-token generation task. We conduct initial supervised fine-tuning on our constructed dataset of 458K samples derived from existing GUI operation datasets. Then we perform iterative refinement using 59K process-reward samples collected during self-improvement cycles. For the reward model training, we use the AdamW optimizer with a learning rate of 1e-5 and a global batch size of 160. We fully fine-tune the model parameters while keeping the vision encoder frozen. Input images are constrained to a maximum of 602,112 pixels to maintain computational efficiency.

### D.3.2 Agent Model Training

Our agent model training follows a progressive enhancement approach:

**Initial model establishment:** We first train a 7B baseline agent using existing datasets from Android-Control, AMEX, and AndroidLab. This model is fully fine-tuned with only the vision encoder frozen, using a learning rate of 1e-5 and batch size of 160.

**Iterative self-improvement:** Using this initial agent, we conduct three rounds of trajectory exploration in dynamic Android environments under UI-Genie-RM's supervision. This process generates UI-Genie-Agent-16K, our synthetic dataset containing 16K high-quality trajectories without manual annotation. After each round, we continue fine-tuning the 7B model on newly collected trajectories for one epoch, maintaining the same hyperparameters.

**Model scaling:** Finally, we train UI-Genie-Agent at different parameter scales (3B and 72B) using the combined UI-Genie-Agent-16K and open-source datasets. The 3B model is fully fine-tuned with frozen vision encoder using AdamW (lr=1e-5, batch size=160) We fine-tune the 72B model using rank-stabilized LoRA (rslora) due to GPU memory constraints, with lora modules added to the vision encoder, projection layer, and LLM. We set both lora rank and lora alpha to 64 and 256 respectively, and initialize lora weights using the PiSSA method.

This progressive training approach enables us to effectively leverage both existing and synthetic data while efficiently scaling our model across different parameter sizes.

# E    Prompt for Evaluation

In this section, we provide the detailed prompts used to adapt our comparative baseline models for reward evaluation tasks. Since neither proprietary MLLMs (GPT-4o, Gemini1.5-pro, Gemini2.5-pro) nor open-source models (Qwen2.5-VL-7B/72B) are specifically trained for reward modeling in GUI operations, we carefully craft specialized prompts to enable fair comparison. Our step-level reward evaluation prompt and outcome-level reward evaluation prompt are presented in Fig. 5 and Fig. 6, respectively. These prompts were designed to provide structured guidance for evaluating both individual action correctness and overall task completion success, leveraging the models' capabilities for processing visual information from UI screenshots.

# F    Example of UI-Genie Training Data

## F.1    Reward Data Example

We present examples of our reward data used for model training. Fig. 7 illustrates action-level reward data samples, which include both positive and negative action examples. These samples are crucial for training our model to distinguish effective UI interactions from ineffective ones. Fig. 8 showcases trajectory-level reward data, comprising complete successful trajectories alongside failed ones, which serve as positive and negative examples respectively.

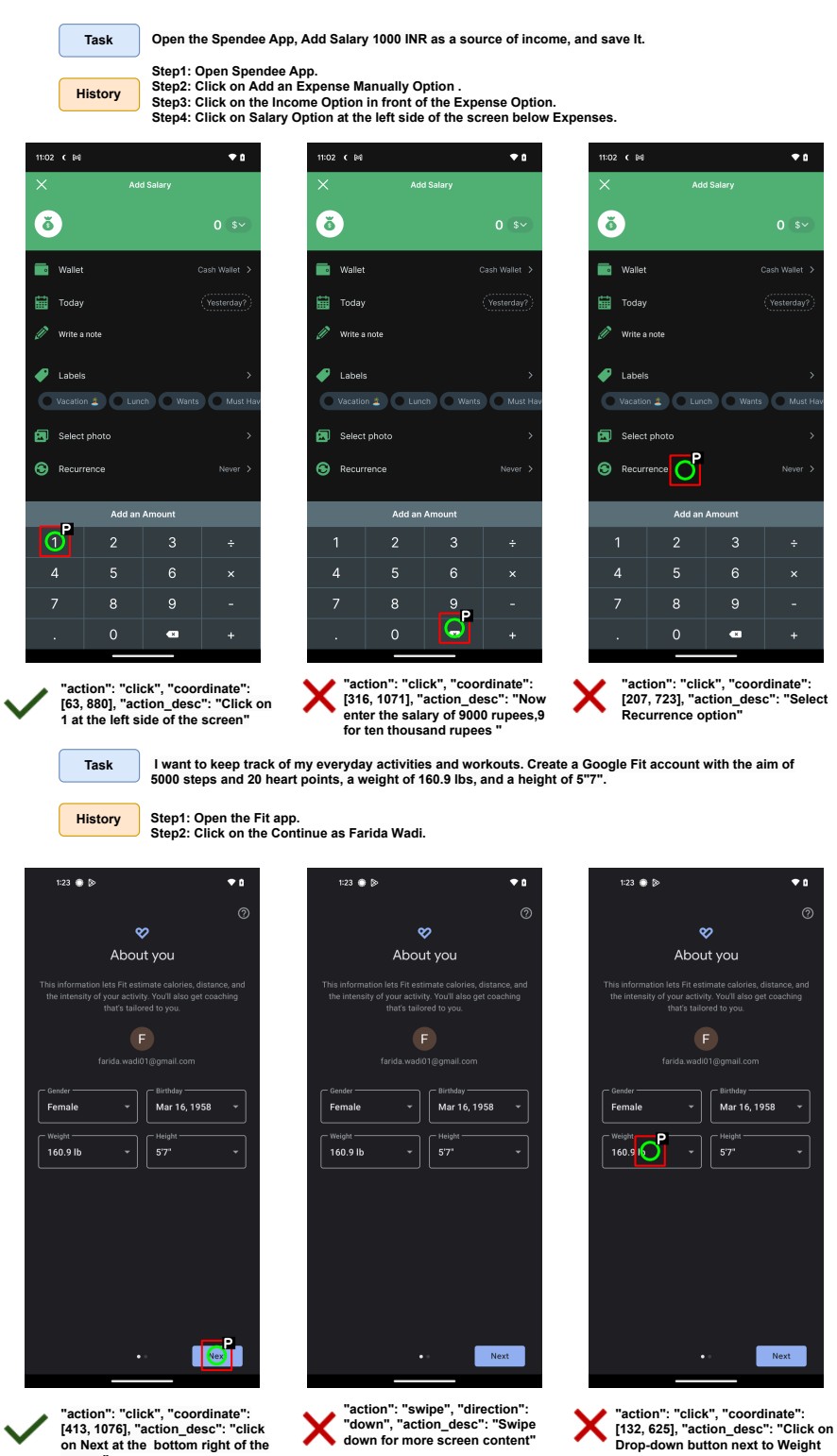

Figure 7: Examples of action-level reward data used for UI-Genie-RM training. The visual prompts displayed here are for illustration purposes only and are not used during model training. For simplicity, we omit the history image displays, though they are included in the actual training process.

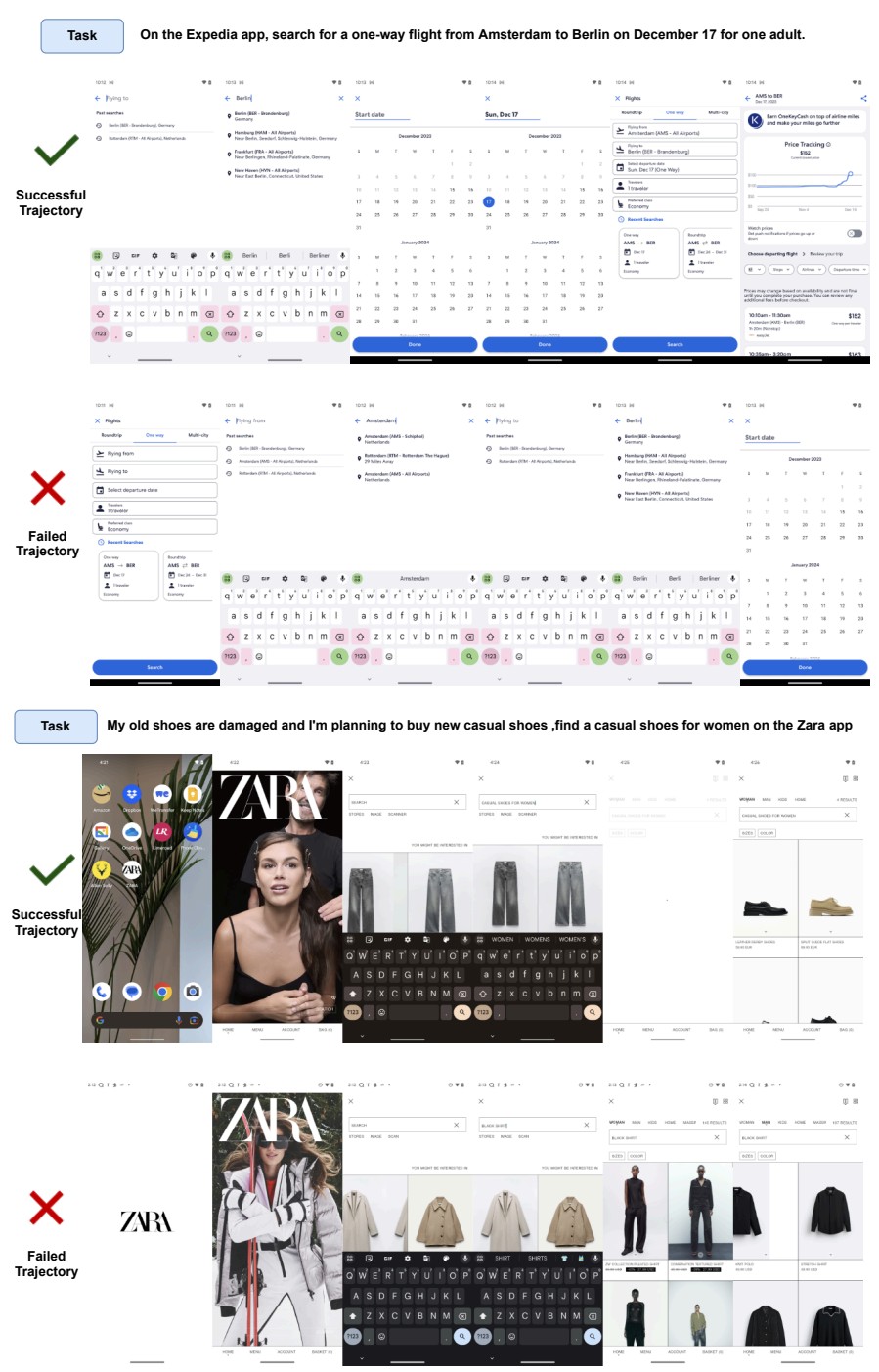

Figure 8: Examples of trajectory-level reward data showing successful and failed task completions. We omit the history of actions in this visualization, but the complete action sequences are utilized during training.

## F.2    Synthetic Trajectory example

We present illustrative examples of trajectories synthesized by UI-Genie, demonstrating both successful and unsuccessful attempts at completing UI tasks. Fig. 9 showcases a successful trajectory generated under the guidance of UI-Genie-RM. As shown in the sequence, the reward model effectively steers the agent through a series of actions, resulting in the successful completion of the assigned task. In contrast, Fig. 10 presents a failure case that illustrates a limitation in trajectory exploration. Although the reward model provides guidance for individual steps within the trajectory, the final outcome reward is negative, indicating overall task failure.

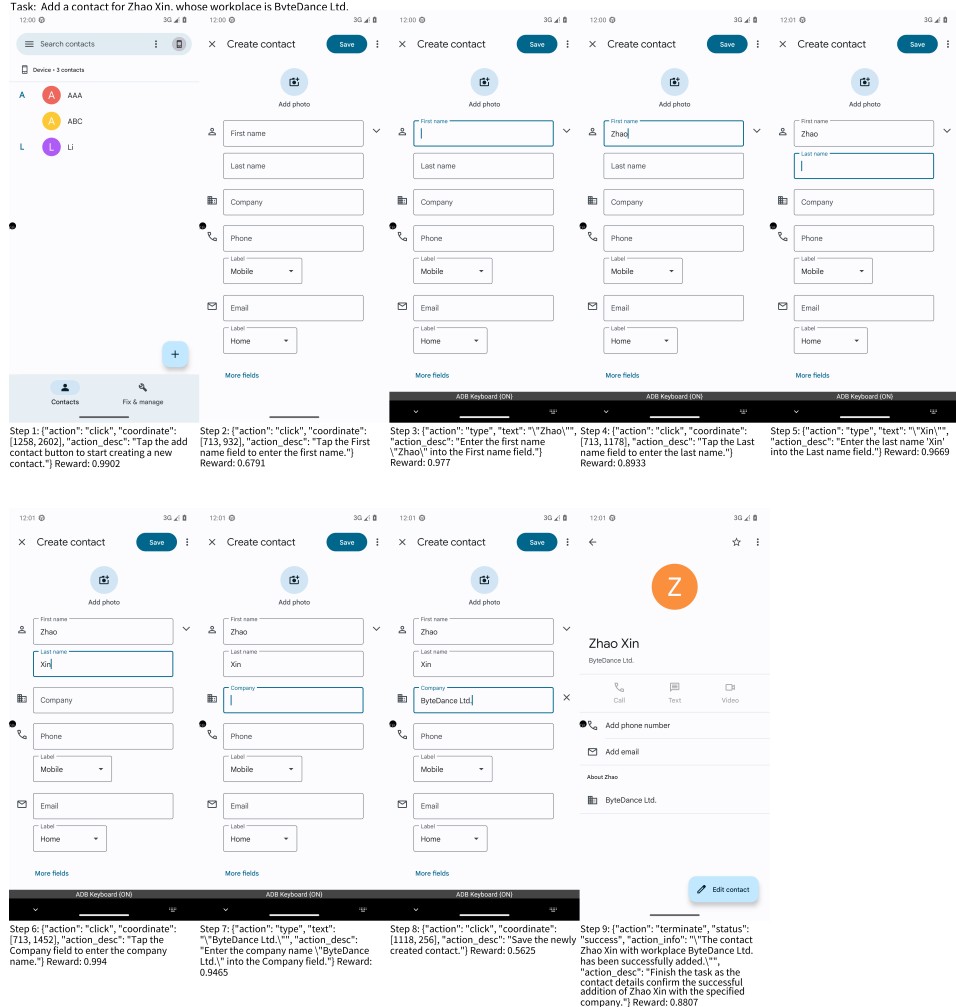

Figure 9: A successful trajectory example. Under the reward guidance of UI-Genie-RM, the agent successfully discovers and executes a sequence of actions that complete the assigned task.

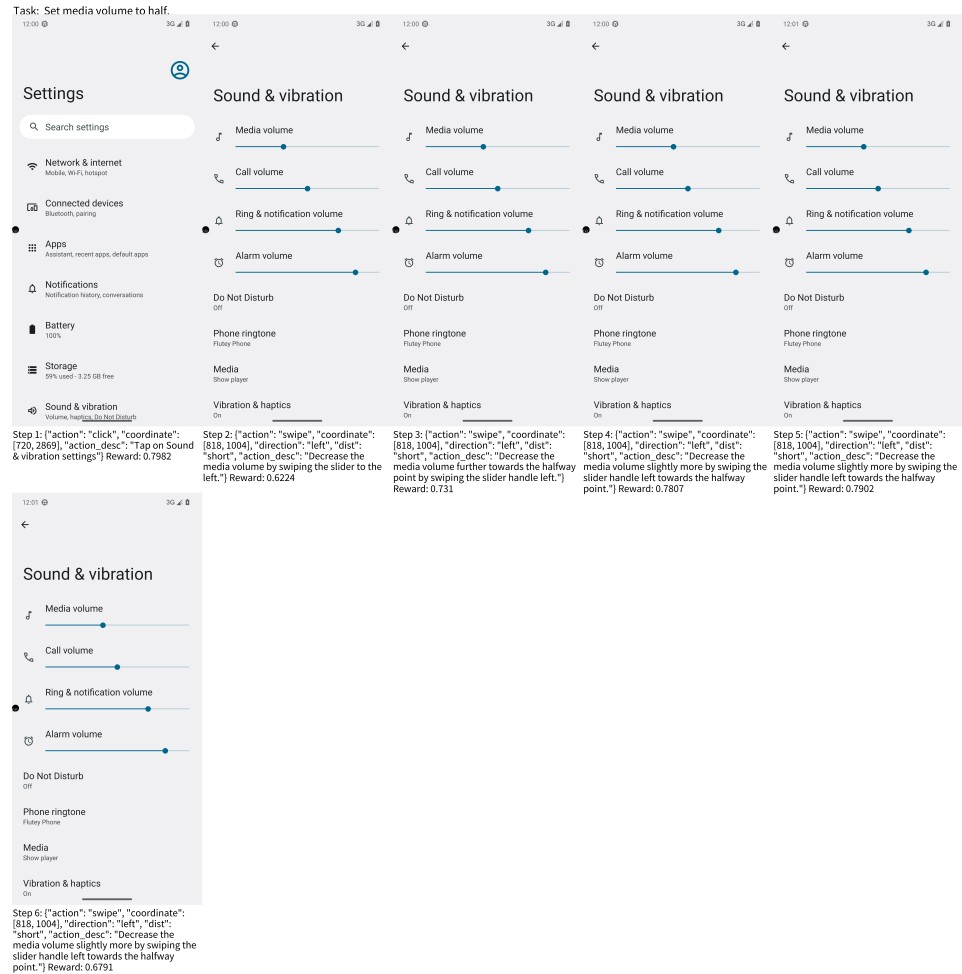

Figure 10: A failure trajectory example. Despite receiving process reward guidance at each individual step, the sequence ultimately results in task failure as indicated by the negative outcome reward.

# G UI-Genie Trajectory Example on Online Evaluation

## G.1 AndroidLab case

We provide examples demonstrating UI-Genie-Agent's capabilities on AndroidLab tasks. Fig. 11 illustrates the agent executing a financial task in which it successfully adjusts an expenditure amount to 500 CNY for the specific date of May 15, 2024 in bluecoins app. This example showcases the model's ability to navigate through the application's interface to locate and modify transaction data according to precise user instructions. In Fig. 12, we present another example where UI-Genie-Agent-72B interacts with the calendar application. The agent demonstrates its capability to edit event details, specifically modifying the end time of an event titled "work" to 7:00 PM. This example highlights the model's proficiency in handling time-based modifications within scheduling applications. These examples from our AndroidLab evaluation illustrate UI-Genie-Agent's ability to understand contextual instructions and execute precise actions within real-world mobile applications, showcasing its versatility across different types of mobile interfaces and task requirements.

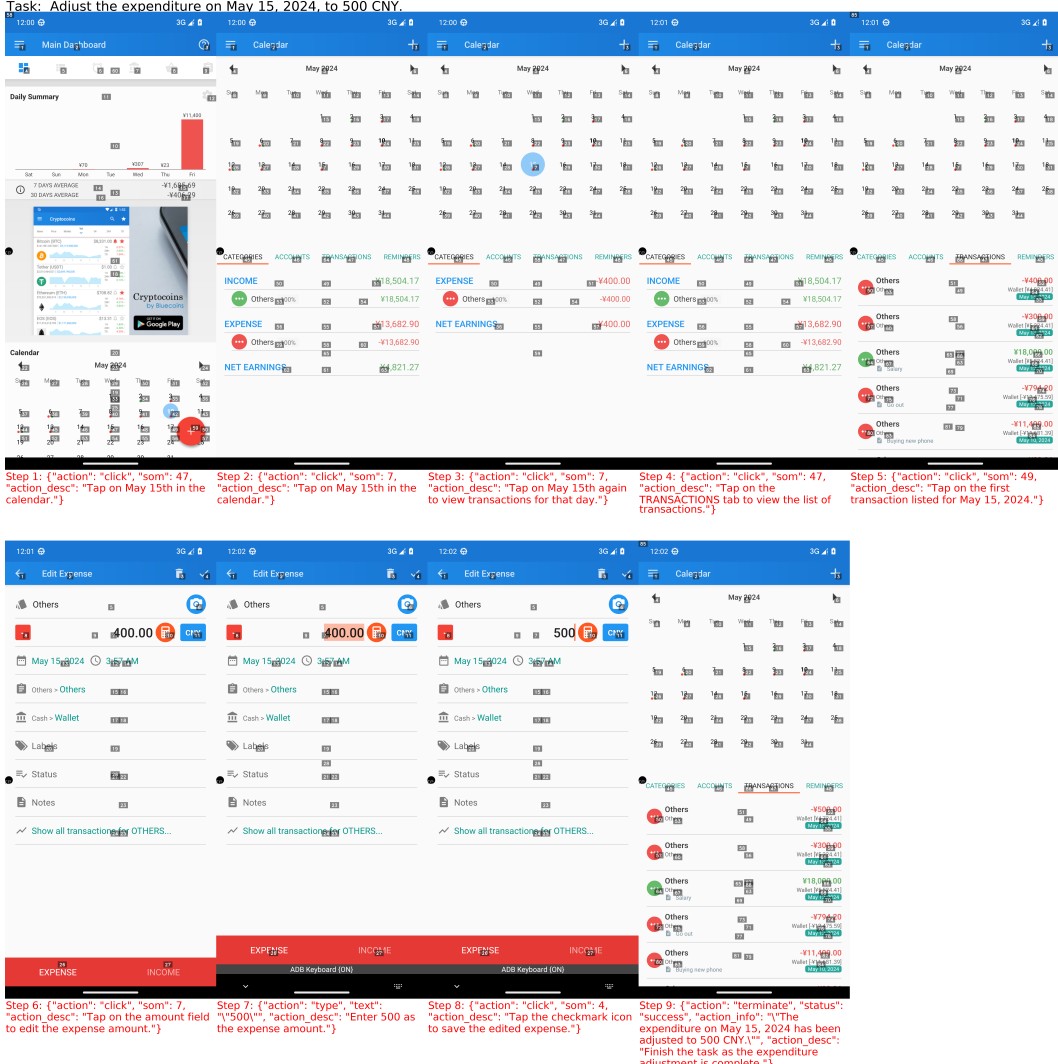

Figure 11: Example of UI-Genie-Agent-72B executing an AndroidLab task. The task instruction: Adjust the expenditure on May 15, 2024, to 500 CNY.

Task: Edit the event with title "work", change the end time to be 7:00 PM.

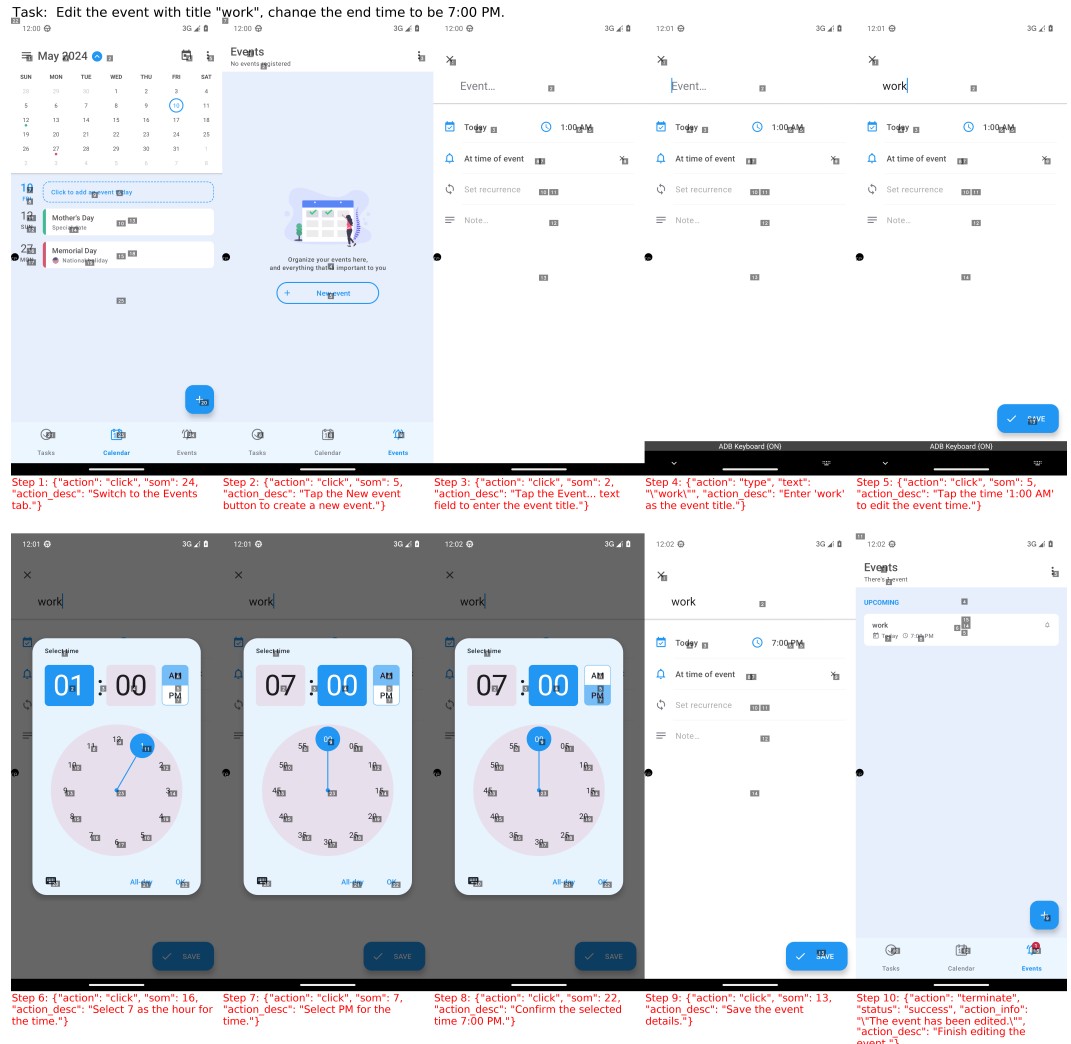

Figure 12: Example of UI-Genie-Agent-72B executing an AndroidLab task. The task instruction: Edit the event with title "work", change the end time to be 7:00 PM.

## G.2  Android Arena (A3) case

We further evaluate UI-Genie's capabilities using the Android Arena (A3) benchmark. Fig. 13 demonstrates UI-Genie-Agent-7B executing a news search task within the CNN application. The agent successfully interprets the task instruction and navigates through the application's interface to locate and utilize the search functionality, ultimately retrieving panda-related news articles. This example highlights the model's ability to understand domain-specific instructions and operate effectively within media consumption applications. Fig. 14 presents another example of UI-Genie-Agent-7B's versatility, showing the agent completing a settings modification task in the Coursera application. The agent demonstrates its capacity to navigate through application menus, locate configuration options, and toggle specific settings. These examples illustrates UI-Genie-Agent's robust performance across diverse applications and task types, from content searches to application configuration changes.

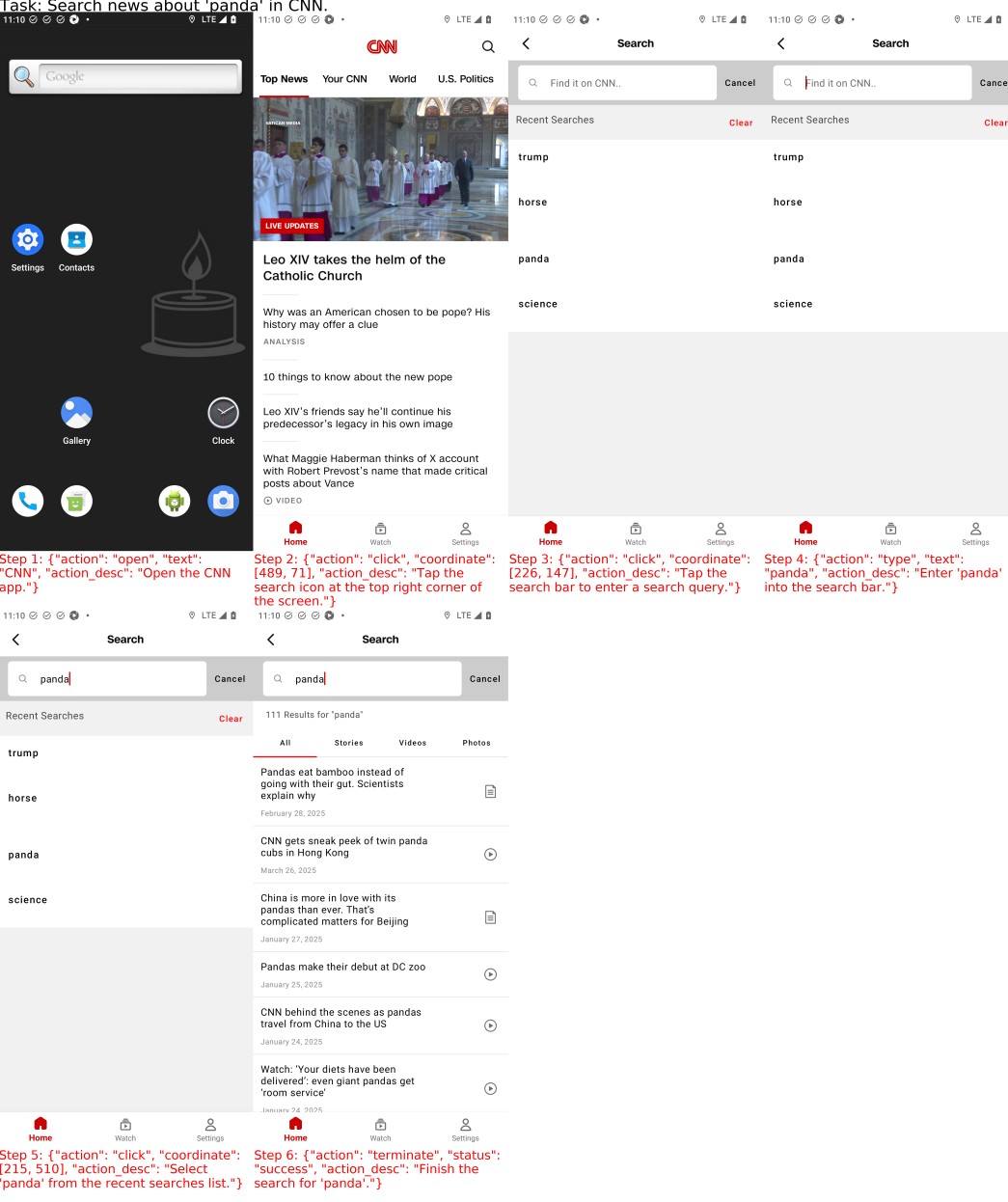

Figure 13: Example of UI-Genie-Agent-7B executing a task defined in Android Arena (A3). The task instruction: Search news about 'panda' in CNN.

Task: Open 'settings' in Coursera and switch to dark mode.

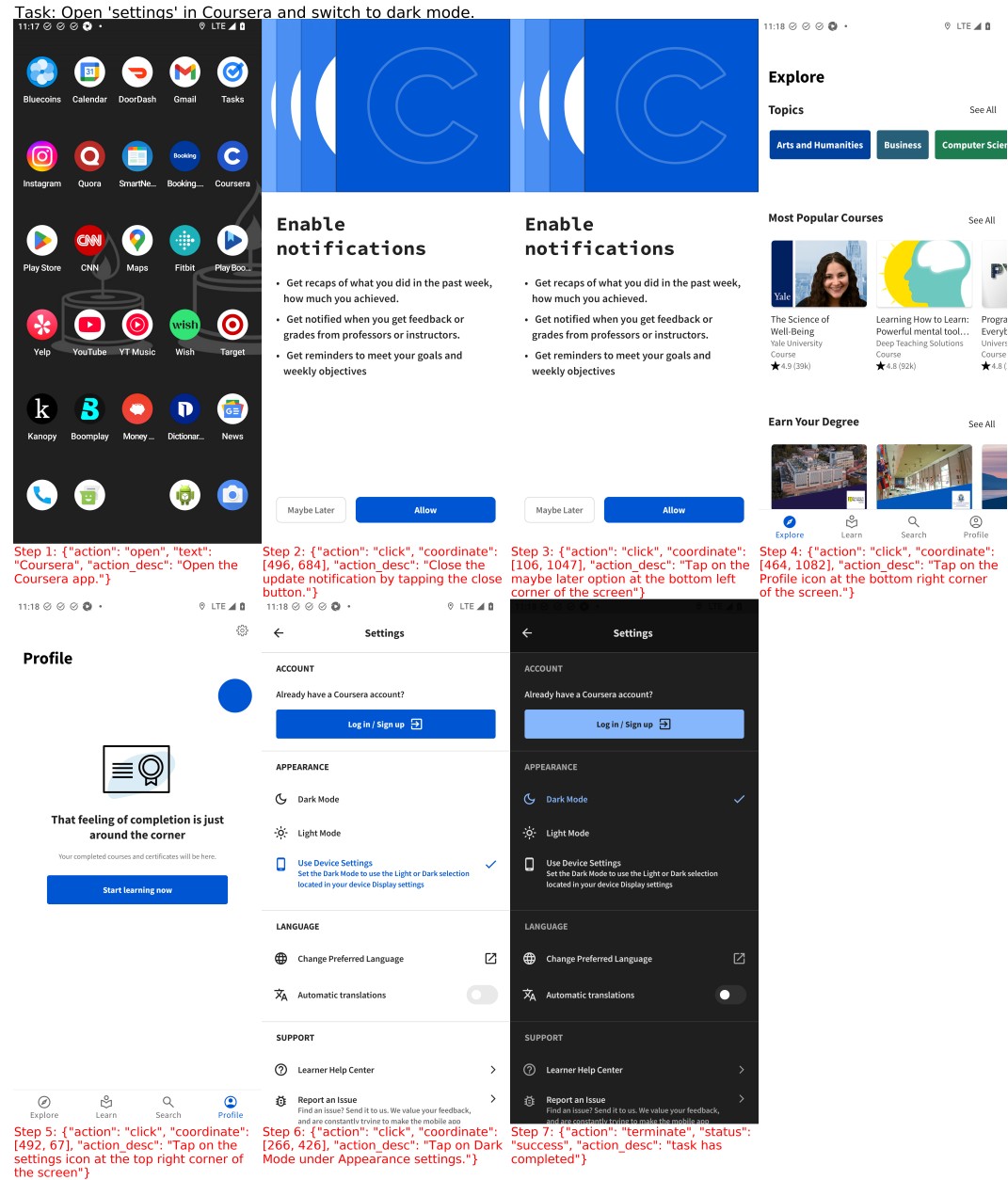

Figure 14: Example of UI-Genie-Agent-7B executing a task defined in Android Arena (A3). The task instruction: Open 'settings' in Coursera and switch to dark mode.

