# OpenReview forum: "UI-Genie: A Self-Improving Approach for Iteratively Boosting MLLM-based Mobile GUI Agents"
_NeurIPS.cc/2025/Conference — NeurIPS 2025 poster_

### Official Review · Reviewer_afLe · 2025-07-01

**Clarity:** 2
**Significance:** 3
**Originality:** 4
**Rating:** 4
**Confidence:** 3

**Summary:**

This paper introduces UI-Genie, a self-improving framework designed to enhance the capabilities of mobile GUI agents by addressing two primary challenges: the difficulty of verifying task outcomes and the scalability issues of creating high-quality training data. To tackle these problems, the authors developed UI-Genie-RM, a specialized reward model that assesses agent performance at both the individual action and full trajectory levels using an image-text interleaved architecture to process historical context. To train this reward model without costly human annotation, the framework employs deliberate data generation strategies, including rule-based verification, trajectory corruption, and hard negative mining. The core of the contribution is a self-improvement pipeline where the agent and reward models evolve iteratively; the reward model guides the agent's exploration in dynamic environments, and the resulting successful or informative failed trajectories are used to expand the training datasets for both models, enabling them to solve increasingly complex tasks. Through this process, the authors created two new datasets, UI-Genie-RM-517k and UI-Genie-Agent-16k, and demonstrated that their UI-Genie agent achieves state-of-the-art performance across multiple benchmarks.

**Questions:**

1. Confusions stem from reading Section 3
- Why rule-based verification is needed for constructing data for the reward model? Will it filter out some data from the base GUI operation trajectory datasets?
- In the iterative model fine-tuning, what is the definition of "rounds", "steps"?
- UI-Genie-Agent-16k is not mentioned in Section 3 but is it generated during the iterative improvement? How?


2. The iterative improvement seems only produces 59k data which is less than 15% of the total training data for the proposed reward model. How does this iterative process better than just using the trained RM as a guide to collect training data and fine-tune the GUI agent?

**Ethical Concerns:**

["NO or VERY MINOR ethics concerns only"]

**Final Justification:**

The paper introduces a novel and impactful self-improving framework for mobile GUI agents, along with a dedicated reward model and new datasets. After reading the rebuttal and discussion, several concerns were satisfactorily addressed:

- Historical context handling is supported by thorough ablations showing the 5-screenshot window balances performance and efficiency.
- Section 3 clarifications—on rule-based verification, dataset generation, and iterative fine-tuning—were clearly explained.
- Iterative improvement effectiveness is validated through performance gains across rounds (Table 3).
- Out-of-domain generalization is demonstrated with UI-Genie-RM performing competitively on unseen apps (Table 4).

Remaining concern is only that the writing and presentation especially in Section 3 remain somewhat hard to follow and could be improved.

**Limitations:**

yes

**Quality:**

3

**Strengths And Weaknesses:**

**Strengths**

1. The paper introduces a novel self-improving framework that iteratively enhances both an agent and a reward model.

2. The development of UI-Genie-RM is the first reward model specifically designed for evaluating GUI agent trajectories, providing a sophisticated solution to the complex problem of trajectory verification.

3. UI-Genie demonstrates superior performance across multiple static and dynamic benchmarks, including AndroidControl, AndroidLab, and A3.

4. The authors plan to open-source the entire framework, including the two novel datasets (UI-Genie-RM-517k and UI-Genie-Agent-16k) and model implementations. This will be a valuable resource for the research community and will undoubtedly facilitate future advancements in GUI agents.

**Weakness**

1. The model processes historical context by using only the five most recent screenshots and summarizing earlier steps as text. It may be insufficient for complex tasks requiring long-term memory while being unnecessarily inefficient for simpler tasks.

2. Section 3 includes some vague descriptions without further explanation or examples, making it hard to follow. See some questions in the following Questions section.

3. While the paper proposed a self-improvement pipeline with three iterative rounds, the experiments does not further analyze or quantify the performance gains from this pipeline.

4. The experiments do not include studies on the out-of-domain performance of UI-Genie-RM.

---

> ### Author Rebuttal · Authors · 2025-07-31
>
> Thank you for your comprehensive review and supportive feedback on our paper. We appreciate your recognition of the novelty of our self-improving framework, the specific contribution of UI-Genie-RM as the first specialized reward model for the GUI agent domain, the superior performance across multiple benchmarks, and the value of our open-source commitment. Below we address specific points raised in your review.
>
> **W1: The sufficiency of using five most recent screenshots to process historical context.**
>
> **A1:** We conduct an ablation study analyzing our model's performance and efficiency across different task complexities.
>
> For simple tasks(<5 steps), the actual input is naturally limited by the number of all available history(e.g., 3 screenshots at step 4). We evaluate preserving at most 1, 3, and 5 screenshots in the following Table.1. While using one screenshot is most efficient, it shows limited performance due to insufficient historical context, particularly in outcome reward evaluation. Using up to five screenshots achieves the best performance without introducing significant inference overhead, since the actual number of processed images remains low.
>
> Table.1 Model performance on simple task with different seen images.
> | Img Num | Step-level Acc( %) $\uparrow$ | Outcome Acc( %) $\uparrow$ | Inference Time(s/iter) $\downarrow$ |
> | --- | --- | --- | --- |
> | 1 | 76.4 | 74.5 | 0.59 |
> | 3 | 78.2 | 79.8 | 1.05 |
> | 5 | 83.5 | 84.3 | 1.57 |
>
> For complex tasks(≥8 steps), we evaluate the screenshot number at 5, 8, and 10. We observe that using 5 screenshots achieve comparable performance to 8 screenshots, while increasing to 10 leads to performance degradation, as presented in the following Table.2. This suggests that older screenshots beyond the 5 most recent ones may distract the model by less relevant information. Moreover, larger window sizes significantly increase the computational cost. These results demonstrate the effectiveness of our choice of using five most recent screenshots to achieve the best accuracy and efficiency trade-off.
>
> Table.2 Model performance on complex task with different seen images.
> | Img Num | Step-level Acc( %) $\uparrow$ | Outcome Acc( %) $\uparrow$ | Inference Time(s/iter) $\downarrow$ |
> | --- | --- | --- | --- |
> | 5 | 72.9 | 73.5 | 2.46 |
> | 8 | 73.4 | 75.2 | 5.10 |
> | 10 | 69.7 | 71.3 | 7.81 |
>
> **W2&Q1: Clarifications on Section 3.**
>
> **A2:** We apologize for any lack of clarity in Section 3. We provide explanations as follows and will revise our paper accordingly.
>
> > Why rule-based verification is needed for constructing data for the reward model? Will it filter out some data from the base GUI operation trajectory datasets?
>
> When constructing the initial training dataset for the reward model, the rule-based verification method is used for getting negative step-level actions. Existing human-annotated GUI agent datasets (AndroidControl, AMEX, etc.) only provide successful trajectories consisting of action sequences, which are used as the *positive samples* in reward model training dataset. To get the *negative samples*, we provide the task instruction of GUI agent dataset to our initial agent model, let it sample the step-level actions, then rule-based methods are used to label these step-level actions as negative or positive (Line153-160, rule-based verfication). For example, given a sample click("empty_space") and the corresponding positive action click("Send"), our rules can label the sample click("empty_space") as negative.
>
> Thus, the rule-based methods will not filter out the original ground-truth trajectories, since we treat them as correct positive examples. The rule-based methods is only used to get the step-level negative actions.
>
> > In the iterative model fine-tuning, what is the definition of "rounds", "steps"?
>
> **Round**: The "round" refers to one full cycle of our iterative self-improvement framework, consisting of three phases: 1) Trajectory exploration: The agent generates new trajectories using reward-guided search. 2) Data expansion: Verifying the outcomes of these new trajectories, adding the successful ones to the agent training datasets, and expanding the reward model's dataset with all labeld steps. 3) Fine-tuning: Training both the agent and reward model on these expanded datasets. Based on the above definition, our fine-tuning process thus has three rounds (Line206).
>
> **Step:** A "step" refers to a single action taken by the agent within a task trajectory. At each step, the agent generates a candidate action, which is then evaluated by our UI-Genie-RM to produce a step-level reward.
>
> > UI-Genie-Agent-16k is not mentioned in Section 3 but is it generated during the iterative improvement? How?
>
> Yes, the UI-Genie-Agent-16k dataset is directly generated during the iterative improvement loops (Line239). As detailed in Section 3.3 under "Training Data Expansion", we use UI-Genie-RM for outcome verification to identify successful task completions. These verified, successful trajectories are added to the agent's training set, finally resulting in 16k such trajectories.
>
> **W3: Quantifying performance gains of the self-improvement pipeline.**
>
> **A3:**  We have studied the performance evolution of both the reward model and agent model in Section A.3 and Figure 1 of our supplementary. The key findings is summarized in the following Table.3. The results demonstrate that the RM's performance **continuously improves** with each iteration, which in turn **boosts** the agent's task success rate. This consistent improvement can be attributed to two main factors:
>
> - **High-Quality foundation:** The reward model is initially trained on high-quality data derived from human-annotated trajectories. In subsequent rounds, we integrate this base data with newly collected data, improving the model through exploring diverse trajectories while maintaining its core capabilities.
> - **Iterative enhancement:** Through the iterative process, the agent model enhances its ability to tackle more complex tasks, generating high-quality reward training samples with increased difficulty. This iterative learning improves the reward model's discriminative ability in challenging scenarios.
>
> Table.3 Model performances after different rounds.
> | Model | Metric | Round 0 $\uparrow$ | Round 1 $\uparrow$ | Round 2 $\uparrow$ | Round 3 $\uparrow$ |
> | --- | --- | --- | --- | --- | --- |
> | UI-Genie-RM | Reward Accuracy | 69.2 | 73.6 | 75.4 | 79.6 |
> | UI-Genie-Agent | Task Success Rate | 18.1 | 29.8 | 33.3 | 38.7 |
>
> **W4: Out-of-domain performance of UI-Genie-RM.**
>
> **A4:** We evaluate UI-Genie-RM on a out-of-domain test set constructed from the AndroidWorld benchmark, using 11 apps that were unseen during UI-Genie-RM training. As shown in the following Table.4, UI-Genie-RM achieves performance competitive with the powerful proprietary model Gemini2.5-pro and significantly outperforms other open-source MLLMs, demonstrating strong generalization to unseen domain. We will add these results to Section 4.4.2 and Table.5 of our paper.
>
> Table.4 OOD performance of UI-Genie-RM.
> | Model | **Step-level Acc(%)** $\uparrow$ | **Outcome Acc(%)** $\uparrow$ |
> | --- | --- | --- |
> | Gemini2.5-pro | 71.5 | 75.7  |
> | Qwen2.5-VL-7B | 46.8  | 64.4 |
> | Qwen2.5-VL-72B | 54.2 | 68.0 |
> | **UI-Genie-RM** | **74.5** | **76.2** |
>
> **Q2: Advantage of iteratively improving the reward model.**
>
> **A:** To demonstrate the effectiveness of iterative fine-tuning, we conduct an ablation study where the reward model is not updated after its initial training. As shown in the following Table.5, fixing the RM results in less performance improvement compared with the iterative fine-tuning baseline.The performance gap is most significant in the final round when tackling more complex tasks. This demonstrates that even a relatively small amount of newly generated, high-quality reward data from challenging, dynamic interactions is critical for guiding the agent toward solving harder tasks. We will include these results into our revised paper.
>
> Table.5 Ablation study on reward model fine-tuning.
> | Reward Model Update | Reward Accuracy-Round 0 $\uparrow$ | Reward Accuracy-Round 1 $\uparrow$ | Reward Accuracy-Round 2 $\uparrow$ | Reward Accuracy-Round 3 $\uparrow$ | Agent Success Rate-Round 0 $\uparrow$ | Agent Success Rate-Round 1 $\uparrow$ | Agent Success Rate-Round 2 $\uparrow$ | Agent Success Rate-Round 3 $\uparrow$ |
> | --- | --- | --- | --- | --- | --- | --- | --- | --- |
> | × | 69.2 | 69.2 | 69.2 | 69.2 | 18.1 | 29.8 | 31.2 | 35.5 |
> | √ | 69.2 | 73.6 | 75.4 | 79.6 | 18.1 | 29.8 | 33.3 | **38.7** |

---

> > ### Author Response · Authors · 2025-08-05
> >
> > Dear Reviewer afLe,
> >
> > We sincerely appreciate the time and effort you have dedicated to evaluating our work and providing valuable feedback. In response to your comments, we have provided detailed clarifications to address your specific concerns regarding historical context processing (Table 1-2), Section 3 method clarifications, iterative performance gains (Table 3), out-of-domain evaluation (Table 4), and RM updating benefits (Table 5).
> >
> > As the discussion period will conclude in two days, we would greatly appreciate it if you could kindly take a moment to review our rebuttal at your convenience. Should you have any remaining questions or suggestions, we would be happy to address them promptly.
> >
> > Thank you again for your insightful feedback, which has helped strengthen our work. We look forward to your thoughts.
> >
> > Best regards,
> >
> > The Authors

---

> > > ### Comment · Reviewer_afLe · 2025-08-05
> > >
> > > Thank you for your responses, which helps a lot for a better understanding of the paper.
> > >
> > > I understand that the paper is focusing on mobile GUI agents, however, could you share some insights on if this is potentially feasible for general GUI agents (in web, and os environments)?

---

> > > > ### Author Response · Authors · 2025-08-06
> > > >
> > > > Dear Reviewer afLe,
> > > >
> > > > Thank you for carefully reviewing our rebuttal and for your thoughtful comments. We are delighted that our rebuttal has helped clarify the work, and we will incorporate the clarifications from our rebuttal into the revised paper. We appreciate your insights regarding the potential generalization of our framework to other GUI domains, and we would like to elaborate on how UI-Genie could be adapted for web and os environments.
> > > >
> > > > Other GUI domains such as web or os environments also face the challenges of **developing scalable reward data generation pipeline** and **reliable reward models**. The core component of our UI-Genie framework can be adapted to address these challenges:
> > > >
> > > > - **Data Generation Strategies:** The datapoints of web and os agents share the similar format with our GUI scenario. Thus, our reward data generation methods, including rule-based verification, controlled trajectory corruption, and hard negative mining, can scalably create reward datasets specific to web pages and os systems applications, not just mobile interfaces. With this reward dataset, we can train an initial reward model for web and os scenarios.
> > > > - **Self-Improvement Loop:** Similar to our GUI environment, with the initial agent and reward model, we can adopt the proposed iterative improvement framework to let the agent model explore more complex web / os tasks, then let the reward model filter high-quality task trajectories. These trajectories can be used for finetuning both model. This process can enhance the performance of both agent model and reward model in the interactive setting of web and os environments.
> > > >
> > > > We acknowledge that, it would require significant resources for collecting domain-specific trajectory data and training corresponding models. This makes it difficult to adopt our methods for web / os environments within a short time.
> > > >
> > > > We consider this as a highly promising direction for future research, and we hope our open-source framework will serve as a solid foundation for the community to explore these extensions.
> > > >
> > > > Thank you once again for your insightful engagement and constructive feedback, which has been invaluable in helping us strengthen our work and consider its broader impacts.

---

> > > > > ### Comment · Reviewer_afLe · 2025-08-07
> > > > >
> > > > > Thanks for the response. I have no further questions.

---

> > > > > > ### Author Response · Authors · 2025-08-07
> > > > > >
> > > > > > Dear Reviewer afLe,
> > > > > >
> > > > > > Thank you for the time and effort you have devoted to reviewing our work. Your insightful feedback and guidance are extremely valuable to us.
> > > > > >
> > > > > > Thank you once more for your commitment in evaluating our work.
> > > > > >
> > > > > > Best regards,
> > > > > >
> > > > > > The Authors

---

### Official Review · Reviewer_Q1Jb · 2025-07-01

**Clarity:** 4
**Significance:** 3
**Originality:** 2
**Rating:** 5
**Confidence:** 4

**Summary:**

This paper introduces a framework for train GUI agents. The method allows to automatically generate data, and fine-tune an agent. The trained agent can be used to generate new data, allowing for a cycle of automatic improvements. The pipeline is composed by two models, a reward model for trajectory verification and the policy. They collected a dataset following their methodology, and train models of different sizes. They aim to open-source the framework.

**Questions:**

- Can you provide an analysis of the dataset as well? What are the type of actions there? How is the distribution of goals?
- There are relevant works in web agents space: NNetNav (https://arxiv.org/abs/2410.02907) and inSTA (https://arxiv.org/abs/2502.06776).
- Will you release the datasets and the models as well? Also, be sure to release the framework before the camera-ready, as it affects significance.

**Ethical Concerns:**

["NO or VERY MINOR ethics concerns only"]

**Final Justification:**

The paper studies an important aspect of training a foundation model for GUI agents. The paper is technically solid and well written; the results are promising. The rebuttal didn't change my assessment of the paper, which is to accept it due to its relevance.

**Limitations:**

I will add a sentence about potential negative impact of fully automated bots for devices, for example in ease frauds.

**Paper Formatting Concerns:**

I don't see any problem in formatting.

**Quality:**

3

**Strengths And Weaknesses:**

The paper is well written, and shows promising results. They tested on different architecture sizes and against many models. The objective to automatizing data collection and allow self-improving agents is significant. They claim they will release the framework open-source, which will increase significance. I suggest to also release the dataset, and trained models with it. The work is original, hower, they may add some citations (see questions). The experiments shows interesting results, but I would add an analysis on the diversity of the dataset, which is often a problem with automatic generated datasets.
The paper claim that the methodology is fully-automated, however it seems to need humans for finding hard-negatives in the RM train.

---

> ### Author Rebuttal · Authors · 2025-07-31
>
> Thank you for your insightful review and strong support for our work. We greatly appreciate your acknowledgement of the originality of our framework, the significance of automated data generation and the promising results across various models. Below are our detailed responses to the points your raised.
>
> **W1: Analysis on the diversity of the generated dataset.**
>
> **A1:** We provide the detailed statistics of the UI-Genie-Agent-16k dataset in the following three tables. We will include these results in our revised paper.
>
> **Action Distribution:** Our dataset contains 8 distinct action types with a realistic distribution of GUI interactions. While Click is naturally the most frequent action, the significant presence of Type (9.7%) and Swipe (6.7%) ensures the agent learns to handle complex data entry and navigation tasks.
>
> Table.1 Analysis on action distribution.
> | Action Type | Open | Click | Swipe | Long press | Type | System button | Wait | Terminate |
> | --- | --- | --- | --- | --- | --- | --- | --- | --- |
> | Number | 2208 | 8428 | 1054 | 15 | 1526 | 218 | 25 | 2208 |
> | Percentage | 14.08% | 53.74% | 6.72% | 0.10% | 9.73% | 1.39% | 0.15% | 14.08% |
>
> **Goal Diversity:** We categorize task goals into 7 distinct categories, showing significant diversity across mobile GUI scenarios. The task goals span a wide range of realistic user intents, preventing model overfitting to a narrow set of tasks.
>
> Table.2 Analysis on goal category.
> | **Goal Category** | Number | **Percentage** |
> | --- | --- | --- |
> | Configuration & Settings | 518 | 23.46% |
> | Information Retrieval | 610 | 27.63% |
> | Task & Item Management | 291 | 13.18% |
> | Search Tasks | 194 | 8.79% |
> | Application Control | 178 | 8.06% |
> | Site Navigation | 183 | 8.29% |
> | Data Entry | 234 | 10.60% |
>
> **Task Complexity:** Crucially,  our dataset emphasizes challenging tasks over trivial simple tasks. We find that medium (5-10 steps) and hard tasks (>10 steps) comprise over 91% of the dataset, which is essential for developing highly capable agents.
>
> Table.3 Analysis on task difficulty.
> | Task Difficulty | Trajectory Length | Number | Percentage |
> | --- | --- | --- | --- |
> | Easy | <5 Steps | 182 | 8.24% |
> | Medium | 5-10 Steps | 1165 | 52.76% |
> | Hard | >10 Steps | 861 | 38.99% |
>
> **W2: Lack of some citations.**
>
> **A2:** We appreciate this feedback and will incorporate these relevant work in our revision.
>
> NNetNav proposes an unsupervised approach for training brower agenets by generating synthetic demonstrations through exploration and retroactive trajectory labeling. InSTA presents an internet-scale dataset construction pipeline that use LLMs to automatically generate agentic tasks, complete these tasks, and filter successful trajectories without human supervision. Both work exploit automatic data generation and trajectory filtering mechinisms, aligning with out UI-Genie's self-improving pipeline conceptually. However, our approach addresses mobile GUI tasks with visual information and introduce a specialized reward model UI-Genie-RM for both action-level and task-level trajectory assessments.
>
> **W3: Open-source release of framework, datasets, and trained models.**
>
> **A3:** We fully agree with your suggestion regarding open-source releases. To maximize the impact and significance of our contribution, we commit to releasing:
>
> - **Complete Framework**: The entire UI-Genie framework including data generation, training, and evaluation pipelines.
> - **Datasets**: UI-Genie-RM-517k (reward model training data) and UI-Genie-Agent-16k (agent model training data).
> - **Trained Models**: UI-Genie-RM (reward model) and UI-Genie-Agent in all tested sizes (3B/7B/72B).
>
> **W4: Human efforts needed for finding hard-negatives.**
>
> **A4:** Our hard-negative mining process is fully automated and requires **no** human effort.
>
> After obtaining the initial agent model (trained on existing trajectory datasets, Line153-154 main paper) and reward model (whose training dataset is constructed with *rule-based verfication* and *controlled trajectory corruption* techniques), the hard-negative mining process is adopted to find challenging negative examples in reward dataset.
>
> We start with existing annotations from the AMEX dataset which provides task target and ground-truth actions. It also provides functional descriptions for other **target-unrelated** UI elements. We then use an open-source LLM (DeepSeek-v3) to convert these descriptions into executable actions. These actions are all regarded as step-level negative actions for the task target. After that, we use the initial reward model to score them. If the reward model mistakenly assign a positive score to one action, this action is identified as hard negative. This process is fully automatic without human annotations. We will revise our description in the paper to make this automated pipeline more clear.
>
> **Response to "Limitations"**
>
> **Potential negative impact of fully automated bots for devices**
>
> **A:** Thank you for highlighting this critical consideration. While the UI-Genie framework aims to advance automation in GUI interactions, we acknowledge the importance of recognizing potential risks associated with fully automated bots. If not properly monitored, these bots could lead to misuse, such as conducting financial fraud or stealing private data without authorization. We will expand the **Broader Impact** section of our paper to include a direct discussion of these risks.

---

> > ### Author Response · Authors · 2025-08-05
> >
> > Dear Reviewer Q1Jb,
> >
> > We sincerely appreciate the time and effort you have dedicated to evaluating our work and providing valuable feedback. In response to your comments, we have provided detailed clarifications to address your specific concerns regarding dataset diversity analysis (Table 1-3), adding suggested citations, open-source commitment, and fully-automated hard-negative mining clarification.
> >
> > As the discussion period will conclude in two days, we would greatly appreciate it if you could kindly take a moment to review our rebuttal at your convenience. Should you have any remaining questions or suggestions, we would be happy to address them promptly.
> >
> > Thank you again for your insightful feedback, which has helped strengthen our work. We look forward to your thoughts.
> >
> > Best regards,
> >
> > The Authors

---

> > ### Comment · Reviewer_Q1Jb · 2025-08-07
> >
> > Thank you for the complete rebuttal, and apologies for my late response. Your responses effectively address all concerns raised.
> > The dataset statistics demonstrate diversity across actions, goals, and task complexity. I am still concerned about the scalability of hard negatives, as this is limited by the initial dataset (AMEX), and cannot scale on synthetic data only. However, this doesn't change my positive assessment of your contribution; I will maintain my positive score.

---

> > > ### Author Response · Authors · 2025-08-07
> > >
> > > Dear Reviewer Q1Jb,
> > >
> > > Thank you for your thoughtful review and for acknowledging our responses. We sincerely appreciate your insightful feedback and guidance, which are invaluable to us.
> > >
> > > Regarding your concern about the scalability of hard negatives, we agree that this is an important consideration. While our current approach leverages the AMEX dataset to construct hard negatives for training the initial reward model, we have found that this relatively small set of high-quality hard negatives—when combined with our self-improvement framework—can yield strong performance. However, we fully recognize the limitations of relying solely on AMEX annotations for scalability, and we deeply appreciate your insight on this point. Your feedback has inspired us to explore approaches for expanding hard negative sampling, such as leveraging our generated trajectory datasets. We view this as a promising direction for future work.
> > >
> > > Once again, thank you for your time and dedication to evaluating our work. We are particularly grateful for your valuable suggestion.
> > >
> > > Best regards,
> > >
> > >  The Authors

---

### Official Review · Reviewer_grio · 2025-07-02

**Clarity:** 2
**Significance:** 3
**Originality:** 2
**Rating:** 4
**Confidence:** 4

**Summary:**

The paper presents UI-Genie, a self-improving framework for GUI agents that tackles two main challenges: (1) difficulty in verifying task outcomes, and (2) lack of scalable high-quality training data. To solve the first, it introduces UI-Genie-RM, a reward model with an image-text interleaved architecture that unifies action- and task-level feedback. It leverages strategies like rule-based verification, controlled trajectory corruption, and hard negative mining for synthesizing reward model training data. Then, UI-Genie uses a self-improvement pipeline that iteratively refines both the agent and reward model via reward model-guided exploration. The approach achieves state-of-the-art performance on multiple GUI benchmarks.

**Questions:**

**Equation 1 Typo**: There may be a typo in Equation 1. The two '-' symbols look questionable. Should the first one be a '+' and the second a '-'? A clarification here would be helpful.

**Iterative Fine-tuning**: Does the iterative fine-tuning process incorporate any continual learning techniques? Without mechanisms to retain previously learned knowledge, the model may suffer from catastrophic forgetting over time. Clarification on this point would be useful.

**Scope of GUI Evaluation**: The evaluation focuses primarily on mobile GUI tasks. It would be informative to see whether the proposed pipeline generalizes to other GUI domains such as desktop or web environments.

**Ethical Concerns:**

["NO or VERY MINOR ethics concerns only"]

**Final Justification:**

With multiple rounds of discussion, the authors resolved most of my questions. I will raise my score.

**Limitations:**

See Weaknesses part.

**Quality:**

2

**Strengths And Weaknesses:**

## Strengths

**Performance**: The model demonstrates impressive performance across three Android-related benchmarks, outperforming some proprietary models despite its smaller size. This is a strong indicator of the method's efficiency and practicality.

**Writing**: The writing is clear and accessible, and the method is presented in a simple yet effective manner. The motivation for self-improvement especially in the context of GUI tasks is compelling, given the high cost of human annotation in such domains.

## Weaknesses

**Claim About Novelty**: The authors claim this is the first method to train a reward model using open-source foundation models. However, the WebRL paper (published in Nov. 2024) also involves training outcome reward models, although in a different setting. While WebRL may not target multimodal GUI tasks specifically, it seems to share the basic method idea that could generalize to this domain. It would strengthen the paper to include a discussion of this overlap and clarify how the proposed approach differs.

**Negative Sample Construction**: The process for constructing negative samples raises some concerns. The assumption that all non-ground-truth actions are incorrect might not hold in practice. For instance, when increasing the quantity of a product from 1 to 2, a user might either click '+' or directly input '2' and they are both valid actions. Rule-based methods may misclassify one as incorrect. It would be valuable to include a more rigorous human evaluation of the negative sample construction process and explore ways to make it more robust.

**Reward Model Improvement**: The iterative training of the reward model raises questions about its long-term reliability. While the first iteration benefits from high-quality human-annotated data, subsequent rounds rely on predictions from a previous model, which may introduce noise. If the reward model starts to accept false positives as correct, this could significantly degrade the quality of future negative samples. More evidence is needed to support the claim that the reward model continues to improve over iterations.

**Experiments**: More detailed results across iterations for both the policy and reward models would better illustrate the effectiveness of the self-improvement loop. It would also be interesting to study ablations, such as removing complex task generation or the other alternative processes, to understand the contribution of each component. Also, existing works on tree search (Koh et al., 2024) could offer competitive alternatives to beam search. A broader comparison to such baselines would strengthen the experimental section.

---

> ### Author Rebuttal · Authors · 2025-07-31
>
> Thank you for your thorough review and constructive feedback provided on our work. We appreciate your recognition of our method's impressive performance, clear writing, and compelling motivation for self-improvement in the costly domain of GUI annotation. Below are our clarifications and responses to the points raised:
>
> **W1: Novelty of our paper and the relation to other work like WebRL.**
>
> **A1:** We focus specifically on developing the first reward model tailored for GUI agents, addressing unique challenges in this domain that cannot be directly adapted from other areas. We will add a discussion distinguishing our work from WebRL in the revised paper.
>
> **Similarities with WebRL**: Both methods can provide outcome rewards for trajectories, enabling the model to learn from both successful and failed attempts.
>
> **Key distinctions from WebRL:**
>
> - **Scalable reward data generation:** The ORM training data of WebRL relies on existing WebArena-Lite training datasets with task-specific evaluation functions. In constrast, GUI agents lack such data, and justifying GUI task completion requires combining screenshot information, rather than easily applying task-specific evaluation functions. We propose a suite of **deliberately-designed techniques** (rule-based verification, controlled trajectory corruption, hard negative mining) to **create a large-scale reward dataset from scratch**, addressing a fundamental challenge for training GUI RMs.
> - **Fine-grained process rewards:** Unlike WebRL's outcome-only reward, our model provides both **task-level** and **action-level** rewards. This fine-grained feedback enables step-wise correctness evaluation and our novel reward-guided beam search, an efficient exploration strategy not present in other work.
> - **Reward model-guided self-improvement:** Different from WebRL, we utilize our reward model not just for trajectory validation, but for actively generating high-quality trajectories without human annotation. Then, our framework refines both reward and agent models iteratively with the increasingly complex training data. In contrast, WebRL's reward model remains fixed after pre-training, limiting its adaptability and preventing continuous generation of new high-quality data to enhance the reward model.
>
> **W2: Negative sample construction and human evaluation.**
>
> **A2:** In the negative sample construction process, we have established rigorous rules to evaluate the action alignment effectively, rather than a simple `non-GT = negative` assumption. Specifically, as discussed in Section B.1 of the supplementary, besides the basic action types, our action space includes a special parameter: `action_desc`, which describes the target of the action. With the sampled action and `action_desc`, we write the following code to use some pre-defined rule-based functions and potentially trigger an LLM to judge whether the sampled action is a true negative or equivalent to positive:
>
> ```python
> def check_action(pred_action, gt_action, task_instruction):
>     # return True for positive actions, False for negative actions
>     pred_type, pred_param, pred_desc = parse_action(pred_action)
>     gt_type, gt_param, gt_desc = parse_action(gt_action)
>
>     # Rule-based verification
>     if gt_type == 'click':
>         # If types and coordinates match, the predicted action is a positive action.
>         if pred_type == gt_type and judge_coordinate(pred_param, gt_param):
>             return True
>         # If coordinates differ or type is different,
>         # the predicted action might be a positive action (e.g. clicking `+` vs. typing `2`)
>         # it can also be negative (e.g. clicking `+` vs. clicking `-`).
>         # We thus use LLM to judge action equivalence based on `action_desc`.
>         elif pred_type in ['click', 'open', 'type', 'system_button']:
>             return llm_judge_action(task_instruction, gt_desc, pred_desc)
>         else:
>             return False
>
>     # ... other rules for swipe, etc.
> ```
>
>  We provide the following two examples of sampled actions and corresponding judgement results. We will clarify the details and include the examples in the revised paper.
>
> Table.1 Examples of negative sampling result.
> | **Positive Action** | **Positive `action_desc`** | **Sampled Action** | **Sampled `action_desc`** | **Judgement Result** |
> | --- | --- | --- | --- | --- |
> | Click [720, 464] | Tap back arrow to exit. | System Button "Back" | Go back to manage accounts. | Positive |
> | Click [720, 464] | Tap back arrow to exit. | Type"Martin Garrix" |Enter Martin Garrix in the Name text field. | Negative |
>
> **Human evaluation:** We manually check 300 random samples each from AndroidControl, AMEX, and AndroidLab. Our above judgement method achieves **92.2% alignment** against manual labels, confirming the high quality of our generated reward data.
>
> Table.2 Sample type alignment with human evaluation.
> | **Data Source** | **Samples** | **Accuracy** $\uparrow$ |
> | --- | --- | --- |
> | AndroidControl | 300 | 94.3% |
> | AMEX | 300 | 93.0% |
> | AndroidLab | 300 | 89.3% |
> | **Total** | **900** | **92.2%** |
>
> **W3: Validation of continuous improvement of the reward model.**
>
> **A3:** We have studied the performance evolution of the reward model in Section A.3 and Figure.1 of our supplementary. The key findings, summarized in the following Table.3, demonstrate that the RM's performance **continuously improves** with each iteration, which in turn boosts the agent's task success rate. This consistent improvement can be attributed to two main factors:
>
> - **High-quality foundation:** The reward model is initially trained on high-quality data from human-annotated trajectories, and subsequent rounds integrate base data with newly collected data, maintaining core capabilities while exploring diverse trajectories.
> - **Iterative enhancement:** The iterative process allows the agent model to handle more complex tasks and generate high-quality training samples, thus improving the reward model’s discriminative ability in challenging scenarios.
>
> Table.3 Performances of different round models.
> | Model | Metric | Round 0 | Round 1 | Round 2 | Round 3 |
> | --- | --- | --- | --- | --- | --- |
> | UI-Genie-RM | Reward Accuracy $\uparrow$ | 69.2 | 73.6 | 75.4 | **79.6** |
> | UI-Genie-Agent | Task Success Rate  $\uparrow$| 18.1 | 29.8 | 33.3 | **38.7** |
>
> **W4: More experimental results and ablation studies.**
>
> **A4:** We provide experimental results of iterative fine-tuning and more ablation studies as follows.
>
> **Iterative performance**: We provide a detailed iteration-by-iteration analysis in the above Table.3. The results demonstrate substantial and consistent improvement for both models, validating the effectiveness of the self-improvement loop.
>
> **More ablation studies:** To assess the contributions of each component, we conduct ablations on the following two aspects, as shown in Table. 4:
>
> (1) **Removing complex task generation**. Without introducing challenging tasks in the final round, the agent’s improvement is limited after Round 2 (31.2→32.6), highlighting the importance of a curriculum of increasing difficulty.
> (2) **Freezing reward model updates.** Updating the reward model during the self-improvement phase provides essential guidance for trajectory exploration, resulting in higher-quality synthetic data generation and enhanced performance of both models.
>
> Table.4 More ablation studies.
> | Complex Task Generation | Reward Model Update | Reward Accuracy-Round 0 $\uparrow$| Reward Accuracy-Round 1 $\uparrow$| Reward Accuracy-Round 2 $\uparrow$| Reward Accuracy-Round 3 $\uparrow$| Agent Success Rate-Round 0 $\uparrow$| Agent Success Rate-Round 1 $\uparrow$| Agent Success Rate-Round 2 $\uparrow$| Agent Success Rate-Round 3 $\uparrow$|
> | --- | --- | --- | --- | --- | --- | --- | --- | --- | --- |
> | × | × | 69.2 | 69.2 | 69.2 | 69.2 | 18.1 | 29.8 | 31.2 | 32.6 |
> | × | √ | 69.2 | 73.6 | 75.4 | 77.1 | 18.1 | 29.8 | 33.3 | 34.1 |
> | √ | × | 69.2 | 69.2 | 69.2 | 69.2 | 18.1 | 29.8 | 31.2 | 35.5 |
> | √ | √ | 69.2 | 73.6 | 75.4 | 79.6 | 18.1 | 29.8 | 33.3 | **38.7** |
>
>
> **Discussion with tree search:** Our reward-guided beam search provides a diverse set of both successful and informative failed trajectories. This diversity is crucial for iteratively refining our reward model with various positive and negative examples. By contrast, the tree search method is designed to optimize toward a single best solution, pruning other low-reward trajectories. This would prevent us from collecting reward signals from failed trajectories. We will explore adapting tree search-based techniques in future work.
>
> **Q1: Equation 1 Typo**
>
> **A:** Equation 1 is not a typo. The trainining objective in Equation 1 is to maximize the log-likelihood of the correct label. For Positive samples $\mathcal{A}_t^{+}$, we want to maximize $\log P(y^{+}|\cdot)$, equavelant to minizing $-\log P(y^{+}|\cdot)$. Similarly, for negative samples $\mathcal{A}\_t^{-}$, we want to maximize $\log P(y^{-}|\cdot)$, equavelant to minizing $-\log P(y^{-}|\cdot)$.
>
> **Q2: Avoiding catastrophic forgetting in iterative fine-tuning.**
>
> **A:** We do not implement specialized continual learning techniques. We employ a straightforward and effective data mixing strategy commonly used in VLM training. Specifically, in each fine-tuning round, we combine newly collected data with the data from the previous rounds to train both the reward model and agent model.
>
> **Q3: Extension to other GUI domains.**
>
> **A:** Other GUI domains also need scalable data generation and robust reward models. Our methods can create reward datasets and improve agent performance across web and desktop applications. However, this paper focuses specifically on mobile agents, and extending to other environments would require significant computation resources and time for data generation and model training. We consider this a promising direction for future work.

---

> > ### Author Response · Authors · 2025-08-05
> >
> > Dear Reviewer grio,
> >
> > We sincerely appreciate the time and effort you have dedicated to evaluating our work and providing valuable feedback. In response to your comments, we have provided detailed clarifications to address your specific concerns regarding novelty clarification (vs WebRL), negative sample validation (Table 1-2), iterative performance improvements (Table 3), ablation studies of self-improvement loop (Table 4), and technical details.
> >
> > As the discussion period will conclude in two days, we would greatly appreciate it if you could kindly take a moment to review our rebuttal at your convenience. Should you have any remaining questions or suggestions, we would be happy to address them promptly.
> >
> > Thank you again for your insightful feedback, which has helped strengthen our work. We look forward to your thoughts.
> >
> > Best regards,
> >
> > The Authors

---

> > ### Comment · Reviewer_grio · 2025-08-05
> >
> > I greatly appreciate the authors' thorough responses. I'm now convinced by the novelty and the value of the self-improvement experiments. However, the biggest remaining concern lies in the construction of negative examples.
> >
> > In many tasks, it's inherently difficult to define what constitutes a "negative" action. There are often multiple valid ways to complete a task. For instance, when filling out a profile form, one might start with the name field or with gender information, regardless of which field appears first. In such cases, it’s hard to objectively determine which sequence is “better.” While the authors provide an example solvable by code-based validation, I doubt a bit about whether such rule-based approaches can scale or generalize to more diverse and complex scenarios.
> >
> > I also had initially assumed the reward model loss followed a standard contrastive approach like the Bradley-Terry model, by maximizing the distinction between positive and negative responses. It would be helpful to include more experiments or justification around the choice of loss function.
> >
> > Overall, while I appreciate the clarifications, I may maintain my current score.

---

> > > ### Author Response · Authors · 2025-08-05
> > >
> > > Dear Reviewer grio,
> > >
> > > Thank you for your thoughtful consideration of our responses and for acknowledging the novelty and value of our self-improvement experiments. We agree that defining "negative" actions is difficult, especially when multiple valid action sequences can complete a task. We understand your concern that the rule-based method may not enough to label positive / negative for diverse actions.
> > >
> > > **We want to emphasize that the rule-based method is only used for constructing the initial reward model dataset, and our dataset construction approach naturally mitigates your concern.**
> > >
> > > Specifically, when sampling a action of the n-th step for a target instruction, we provide the agent model with first (n-1) ground truth actions and let it sample multiple different actions for the n-th step. This **teacher forcing** mechanism constrains the agent's output distribution to stay reasonably close to the training data distribution. **As a result, our rule-based method typically does not face positive actions that deviate significantly from ground truth patterns, which makes the rule-based strategy sufficient in this constrained setting.** In your profile form example, the agent would typically generate actions like "fill in the name field with 'John', 'Max', 'Justin' etc" based on the previous information, rather than choosing between differernt field orders, since it is conditioned on the previous ground truth context ("open the form", "tap the first field to enter the name".).
> > >
> > > With this **cold start** dataset, we fine-tune an MLLM letting it able to judge whether an action is correct or not based on the history actions.
> > >
> > > To further improve the effectiveness and stability of the reward model in diverse, complex scenarios, we design the **self-improvement loop** to improve it to handle more diverse scenatios. **It overcomes the limitations of static, rule-based labeling by learning directly from dynamic task outcomes**:
> > >
> > > - **Dynamic trajectory generation:** We provide the agent model with a task instruction and let it sample action sequentially. When sampling a new action, it conditions on **previously sampled actions rather than ground truths.** At each sampling step, we retain the top-5 trajectories with the highest trajectory rewards so far. If any output trajectory successfully completes the task, we label all of its actions as positive (line 200 of the paper). This ensures diversity in the final sampled positive trajectories and their constituent actions. This directly tackles the challenge you've identified: **any alternative, valid path (like filling a form in a different order) is correctly identified and used as high-quality positive data**, even if it deviates from the initial human demonstration.
> > > - **Fine-grained labeling from outcomes, not rules:** The ultimate step-level supervision is created based on the real-world task outcome. For unsuccessful trajectories, we use continuation rollouts to determine if an intermediate step **could have led to success through alternative paths** (line 203 of our paper). This allows us to label actions based on their actual potential to solve the task, rather than static rules.
> > >
> > > This iterative process enhances our model's accuracy and generalization on complex scenarios, as demonstrated by the improved reward accuracy from 69.2% to 79.6% in the above Table 3.
> > >
> > > Regarding the loss function choice, we use a binary classification objective because our goal is correctness evaluation, not preference ranking. In GUI tasks, there might be multiple "correct" actions, but it is difficult and often meaningless to say one is "better" than another. As in the form-filling exmple, inputting name or gender information are both correct actions—our reward model should label both as positive, not rank one over the other. A contrastive loss, as in the Bradley-Terry model, is designed to rank responses, especially aligning them with human preference. In contrast, our binary classification loss assigns a positive label to any action that contributes to task completion. This approach is standard for process supervision in related fields like math reasoning (Let’s Verify Step by Step [1], Math-Shepherd [2]) where step-wise correctness is the primary goal.
> > >
> > > Thank you once again for your insightful feedback, which has helped us clarify the most critical aspects of our work. We hope our responses address your concerns effectively. Please let us know if you have any further questions or require additional clarification.
> > >
> > > [1] Lightman, Hunter, et al. "Let's verify step by step." The Twelfth International Conference on Learning Representations. 2023.
> > >
> > > [2] Wang, Peiyi, et al. "Math-shepherd: Verify and reinforce llms step-by-step without human annotations." arXiv preprint arXiv:2312.08935 (2023).

---

> > > > ### Author Response · Authors · 2025-08-09
> > > > **Respectful Inquiry Before Discussion Deadline**
> > > >
> > > > Dear Reviewer grio,
> > > >
> > > > We sincerely appreciate your time and thoughtful engagement with our work. As the discussion period is concluding soon, we would like to kindly confirm if our clarifications regarding (1) negative sample construction (addressing scalability through rule-based methods and self-improvement) and (2) loss function design (binary classification vs. contrastive) have addressed your concerns.
> > > >
> > > > Your insights have been invaluable in strengthening our paper. If any points remain unclear, we would be happy to address them promptly.
> > > >
> > > > Once again, thank you for your dedication to evaluating our work.
> > > >
> > > > Best regards,
> > > >
> > > > The Authors

---

### Official Review · Reviewer_yoko · 2025-07-03

**Clarity:** 3
**Significance:** 3
**Originality:** 2
**Rating:** 4
**Confidence:** 4

**Summary:**

This paper propose a self-improving framework, named  UI-Genie, designed to address two core challenges in GUI agent development: the difficulty of reliably verifying agent trajectories and the lack of scalable, high-quality training data. The proposed framework introduces UI-Genie-RM, a reward model with an image-text interleaved architecture for precise trajectory assessment, supported by targeted data generation strategies. The authors also establish an iterative self-improvement pipeline, enabling exponential growth in solvable GUI tasks and training data. The framework demonstrates state-of-the-art results on multiple benchmarks.

**Questions:**

1. The paper lacks experiments verifying the necessity of training UI-Genie-RM within the proposed framework. If the reward model were replaced with other MLLMs and the reward model fixed during training (i.e., not updated), could similar performance be achieved?
2. How is the performance in the popular dynamic mobile benchmark AndroidWorld?

**Ethical Concerns:**

["NO or VERY MINOR ethics concerns only"]

**Final Justification:**

It has addressed most of my concerns, and considering the technical innovation of the method, I will maintain my borderline positive score.

**Limitations:**

Yes.

**Quality:**

3

**Strengths And Weaknesses:**

Strengths:
1. The workload of this paper is sufficient, and the motivation makes sense.
2. The dataset collected using the proposed method is large in scale, and the experiments verify the effectiveness of training the agent with the dataset.
3. Both the proposed method and dataset will be open-sourced, which facilitates reproducibility for researchers and can also inspire future work.

Weaknesses:
1. Using the reward model to judge the correctness of the data and use it for subsequent training is a common idea, and the technical innovation of this paper should be further clarified.
2. The newly proposed GUI trajectory reward task should be clearly defined before introducing the method, including the input/output information, task objectives, etc., and it would be best to correspond these directly to the content in Fig. 2.
3. In the experiment of 4.4.2 Reward Model Evaluation, the test samples are drawn from the same source as the training set, meaning they are from the same distribution. The superior performance of UI-Genie-RM may result from overfitting. This setup may not be entirely fair; the test set should be drawn from a distribution different from that of the training set.
4. The paper lacks ablation studies on the three complementary data generation techniques used during the Data Construction for Reward Model Training phase. Such experiments are important to demonstrate the effectiveness of the algorithmic design in improving the quality of the training data. It also lacks ablation on whether the reward model is updated or not.

---

> ### Author Rebuttal · Authors · 2025-07-31
>
> Thank you for your thorough review and positive feedback on our paper. We appreciate your recognition of our motivation, workload, large-scale generated data and open-source commitment. We have carefully considered your valuable suggestions and will address your points below.
>
> **W1: The technical innovation of our paper.**
>
> **A1:** While using reward models is a known paradigm, its application to GUI agents is non-trivial and presents distinct challenges. Unlike tasks with verifiable final answers and well-defined reward datasets (e.g. math problems, code generation), GUI tasks involve dynamic screenshots and historical actions, making it difficult to define standard answers, and create large-scale, accurate training data for specialized GUI reward models. Consequently, there lacks a well-established reward model designed for GUI agents that can effectively assess GUI action quality.
>
> Our core innovations directly address these challenges:
> - **Specialized reward data generation techniques:** We develop automated techniques including rule-based verification, controlled trajectory corruption, and hard negative mining to transform existing GUI agent dataset into high-quality reward training data without manual annotation, enabling us to train an initial GUI reward model.
> - **Co-evolution self-improvement pipeline:** We establish a mutual enhancement loop where the agent and reward model continuously improve through dynamic interaction - the agent generates increasingly complex trajectories, the reward model evaluates them, and both model are continuously refined using the generated feedback data along with the initial data.
>
> These contributions provide an efficient, scalable solution for getting the reward model's training data and training both reward and policy models in GUI environments.
>
> **W2: Better definition of the newly proposed GUI trajectory reward task.**
>
> **A2:** Thank you for this suggestion. We will add a new Section 3.1 to formally define the task and revise Figure 2 to align with these definitions. The new section will read:
>
> 3.1 Task Formulation for GUI Trajectory Reward
>
> **Action-level reward:** Given a task goal $\mathcal{G}$, current screenshot $\mathcal{I}\_t$, and historical observations $\mathcal{O}\_t={(\mathcal{I}\_0,\mathcal{A}\_0), (\mathcal{I}\_1,\mathcal{A}\_1), ..., (\mathcal{I}\_{t-1},\mathcal{A}\_{t-1})}$, the objective is to evaluate whether a candidate action $\mathcal{A}\_t$ constitutes a correct step toward achieving $\mathcal{G}$. The reward model outputs a positive reward ($y=1$) for correct actions and negative reward ($y=0$) otherwise.
>
> **Task-level reward**: This task evaluates whether an entire trajectory successfully completes the goal $\mathcal{G}$. The input consists of all actions and resulting screenshots in the trajectory, and the output is a single reward score indicating overall success or failure.
>
> **W3: Out-of-distribution test of UI-Genie-RM.**
>
> **A3:** We evaluate UI-Genie-RM on a out-of-distribution test set constructed from the AndroidWorld benchmark, using 11 apps that were unseen during UI-Genie-RM training. As shown in the following Table.1, UI-Genie-RM achieves performance competitive with the powerful proprietary model Gemini2.5-pro and significantly outperforms other open-source MLLMs, demonstrating strong generalization to unseen domain. We will add these results to Section 4.4.2 and Table.5 of our paper.
>
> Table. 1 OOD test of UI-Genie-RM
> | Model | **Step-level Acc (%)** $\uparrow$| **Outcome Acc (%)** $\uparrow$|
> | --- | --- | --- |
> | Gemini2.5-pro | 71.5 | 75.7  |
> | Qwen2.5-VL-7B | 46.8  | 64.4 |
> | Qwen2.5-VL-72B | 54.2 | 68.0 |
> | **UI-Genie-RM** | **74.5** | **76.2** |
>
> **W4: Ablation studies on three complementary data generation techniques and reward model updating.**
>
> **A4:** The results in the following Table.2 show the impact of three techniques used to construct reward model training data and whether the reward model should be updated in the self-improvement loops.
> Results of the first four experiments demonstrate each technique provides a significant boost to the reward model's accuracy. The comparision between the fourth and fifth experiments shows that updating the reward model in the self-improvement loops provide a large performance improvement. These results validate the effectiveness of data construction techniques and the design choice of self-improvement framework. We will add these results into the revised paper.
>
> Table.2 Ablation studies on three complementary data generation techniques and reward model updating.
> | Rule-based Verification | Trajectory Corruption | Hard Negative Mining | RM Updating | Step-level Acc (%)$\uparrow$ | Outcome Acc (%)$\uparrow$ |
> | --- | --- | --- | --- | --- | --- |
> | × | × | × | × | 56.6 | 58.9 |
> | √ | × | × | × | 67.2 | 65.3 |
> | √ | √ | × | × | 67.4 | 75.9 |
> | √ | √ | √ | × | 69.2 | 76.7 |
> | √ | √ | √ | √ | **79.6** | **82.1** |
>
> **Q1**: **Necessity of training UI-Genie-RM.**
>
> **A:** To verify the necessity of training and updating a specialized RM, we adopt two variants: (1) using the base Qwen2.5-VL-7B as a fixed RM, and (2) using our initial UI-Genie-RM without updates during self-improvement. As shown in the following Table.3, a general MLLM (Qwen2.5-VL-7B) provides minimal benefit and fails to guide the agent effectively on complex tasks, leading to performance degradation. A specialized designed and fixed in the self-improvement phase reward model is helpful but its effectiveness is relatively limited on complex tasks. Our full framework, where the RM is continuously improved, achieves consistent gains, demonstrating the advantage of a **specialized and continuously improving RM.** We will include these analysis in the revised paper.
>
> Table.3 Ablation studies on reward models.
> | Reward Model | Reward Model Update | Reward Accuracy (%)$\uparrow$ | Agent Success Rate (Round 0)$\uparrow$ | Agent Success Rate (Round 1) $\uparrow$| Agent Success Rate (Round 2)$\uparrow$ | Agent Success Rate (Round 3)$\uparrow$ |
> | --- | --- | --- | --- | --- | --- | --- |
> | Qwen2.5VL-7B | × | 56.6 | 18.1 | 22.5 | 23.2 | 21.7 |
> | UI-Genie-RM | × | 69.2 | 18.1 | 29.8 | 31.2 | 35.5 |
> | UI-Genie-RM | √ | 79.6 | 18.1 | 29.8 | 33.3 | **38.7** |
>
> **Q2: Performance on AndroidWorld benchmark.**
>
> **A:** We evaluate our final agent models on the AndroidWorld benchmark. As shown in the Table.4 below, UI-Genie-Agent-7B achieves a success rate of 36.2%, surpassing strong proprietary models like GPT-4o. Our 72B model shows significant performance gains compared to its base model Qwen2.5-VL-72B, outperforming all other methods. We will incorprate these new evaluation results into our revised paper.
>
> Table.4 Comparisions on AndroidWorld benchmark
> | Models | Agent Success Rate (%) $\uparrow$|
> | --- | --- |
> | GPT-4o | 34.5 |
> | Claude Computer Use | 27.9 |
> | Qwen2.5-VL-7B | 22.0 |
> | UI-TARS-7B | 33.0 |
> | **UI-Genie-Agent-7B** | **36.2** |
> | Qwen2.5-VL-72B | 35.0 |
> | Aguvis-72B | 26.1 |
> | **UI-Genie-Agent-72B** | **47.4** |

---

> > ### Author Response · Authors · 2025-08-05
> >
> > Dear Reviewer yoko,
> >
> > We sincerely appreciate the time and effort you have dedicated to evaluating our work and providing valuable feedback. In response to your comments, we have provided detailed clarifications to address your specific concerns regarding GUI-specific innovation, task definition of GUI trajectory rewards, out-of-distribution evaluation of UI-Genie-RM (Table 1), ablation studies (Table 2-3), and AndroidWorld results (Table 4).
> >
> > As the discussion period will conclude in two days, we would greatly appreciate it if you could kindly take a moment to review our rebuttal at your convenience. Should you have any remaining questions or suggestions, we would be happy to address them promptly.
> >
> > Thank you again for your insightful feedback, which has helped strengthen our work. We look forward to your thoughts.
> >
> > Best regards,
> >
> > The Authors

---

> > > ### Comment · Reviewer_yoko · 2025-08-08
> > > **Official Comment of Reviewer yoko**
> > >
> > > Thanks for the detailed response and additional experiments. It has addressed most of my concerns, and considering the technical innovation of the method, I will maintain my current score.

---

> > > > ### Author Response · Authors · 2025-08-08
> > > >
> > > > Dear Reviewer yoko,
> > > >
> > > > Thank you very much for your thoughtful follow-up and for acknowledging our rebuttal. We're glad that our responses and additional experiments have addressed most of your concerns.
> > > >
> > > > Your insightful guidance has been invaluable in improving our work. We sincerely appreciate your dedication to the review process.
> > > >
> > > > Best regards,
> > > >
> > > > The Authors

---

### Note · Authors · 2025-08-13

Dear Chairs and Reviewers,

We are sincerely grateful for your dedication and insightful feedback throughout the review process. We are encouraged that three out of four reviewers gave positive scores, recognizing our work as a **"novel self-improving framework"** (Reviewer afLe) with **"compelling"** motivation (Reviewer yoko, grio) and **"significant"** objectives (Reviewer Q1Jb). They highlighted our framework's **originality** in developing the first GUI-specific reward model (Reviewer Q1Jb) and its effectiveness in solving the **"complex problem of trajectory verification"** (Reviewer afLe), and our **open-sourcing commitment** (Reviewer yoko, Q1Jb, afLe). Our experimental results demonstrate **"impressive performance"** (Reviewer grio) and **"superior results"** (Reviewer afLe) across multiple benchmarks.

During the rebuttal period, we throughly addressed the concerns and suggestions raised by the reviewers. We clarified GUI-specific challenges and our core contributions, specialized reward data generation and co-evolution self-improvement pipeline. We also added extensive experiments including out-of-distribution evaluation on AndroidWorld, comprehensive ablation studies, and iteration-by-iteration performance improvements. In addition, we provided detailed statistics demonstrating significant diversity across action types, goal categories, and task complexity. We also provided clarifications on formal task definitions and negative sampling validation with 92.2% human evaluation accuracy.

We are grateful that all reviewers acknowledged our detailed rebuttal. For Reviewer grio's remaining questions on negative sample construction and loss function choice, we clarified that our initial rule-based approach utilizes teacher-forcing mechanisms and our self-improvement loop further learns from dynamic task outcomes, and justified our binary classification loss as standard for correctness evaluation in process supervision.

We believe our responses have sufficiently addressed the reviewers' concerns, demonstrating the robustness and signifcance of our work. Thank you once again for your invaluable time and guidance.

Best regards,

The Authors

---

### Decision · Program_Chairs · 2025-09-17

**Decision:**

Accept (poster)

**Comment:**

The paper studies the training pipeline for GUI agents to operate mobile phone interfaces, and contributes two components that address the significant challenges in the domain.
The problem is well-motivated and timely as multi-modal LLM-based agents are being applied to different domains like web browsers, desktop OS and mobile interfaces.
The proposed solution components are described clearly and address the challenges of scalable, reliable trajectory collection & evaluation in a practical and effective manner.
The experiments in the paper left a lot of unanswered questions, and the reviewers initially all rated the paper as borderline as a result.
However, the authors substantially improved the paper with additional experiments, ablations, and analysis during the discussion period.
These include:
- Ablating the different data gathering and reward model updating schemes to isolate individual benefit and assess complementarity.
- Assess the learned reward model out-of-distribution.
- Assess the quality of reward labeling heuristics on the generated data.
- Analyze the diversity of generated data.
- Vary the amount of context used for reward modeling, to study accuracy-vs-latency trade-offs.
- etc.

Along with the several clarifications and discussions (e.g. clearer statement of problem formulation, clearer connections to related work on Web agents, etc.) make the paper substantially stronger and should all be included in the revision.
All the reviewers agree that the paper is above the bar for publication after these changes are incorporated.